

# A protocol for model intercomparison of impacts of Marine Cloud Brightening Climate Intervention

Philip J. Rasch[1], Haruki Hirasawa[1,2], Mingxuan Wu[3], Sarah J. Doherty[1], Robert Wood[1], Hailong Wang[3], Andy Jones[4], James Haywood[5, 4], and Hansi Singh[2]

[1]University of Washington, Seattle, WA, USA
[2]University of Victoria, Victoria, BC, CA
[3]Pacific Northwest National Laboratory, Richland, WA, USA
[4]Met Office Hadley Centre, Exeter, UK
[5]University of Exeter, UK

**Correspondence:** Phil Rasch (philip.j.rasch@gmail.com)

**Abstract.** A modeling protocol is introduced (defined by a series of model simulations with specified model output). The protocol is designed to improve understanding of climate impacts from Marine Cloud Brightening (MCB) Climate Intervention. The model simulations are not intended to assess consequences from a realistic MCB deployment intended to achieve specific climate targets but instead to expose responses produced by MCB interventions in 6 regions with pervasive cloud systems that

are often considered as candidate regions for such a deployment. A calibration step involving simulations with fixed sea surface temperatures is first used to identify a common forcing, and then coupled simulations with forcing in individual regions and combinations of regions are used to examine climate impacts. Synthetic estimates constructed by superposing responses from simulations with forcing in individual regions are considered as a means to approximate the climate impacts produced when MCB interventions are introduced in multiple regions

A few results comparing simulations from 3 modern climate models (CESM2, E3SMv2, UKESM1) are used to illustrate similarities and differences between model behavior and the utility of estimates of MCB climate responses that have been synthesized by summing responses introduced in individual regions. There are substantial differences in the cloud responses to aerosol injections between models, but the models often show strong similarities in precipitation and surface temperature response signatures when forcing is imposed with similar amplitudes in common regions.

## 1 Introduction

It is becoming increasingly apparent that there are enormous consequences to society and nature from rising concentrations of atmospheric greenhouse gases (GHGs). Although scientists have long warned of these consequences, current efforts to limit GHG emissions appear inadequate to prevent large and dangerous climate change (IEAWEO22, 2022, fig 3.2). Several approaches for deliberate "climate intervention" (CI, Mcnutt et al., 2015a, b; National Academies of Sciences, Engineering,

and Medicine, 2021) have been proposed as a way to counteract some global warming impacts while GHG emissions are being reduced. One major class of proposed CIs involve aggressive Carbon Dioxide Removal (CDR) but there are substantial



environmental, technical, and cost challenges to use of CDR at the scale needed to significantly and rapidly reduce global warming. Another class, and the primary consideration here, proposes to make slight changes to the energy entering and leaving the planet. These methods use the physical principle that the planet's average temperature is determined by a balance

between the Sun's energy entering the earth system (warming the planet) and its eventual emission away at longer (infra-red) wavelengths (cooling the planet). This strategy operates to cool the planet by slightly reducing the sunlight reaching Earth's surface or increasing the emission of energy leaving the climate system at the surface. CI strategies that modify Earth's energy budget are often broadly called 'Sunlight Reflection', 'Solar Radiation Modification' (SRM) or solar geoengineering methods, because they are usually intended to reduce incoming energy at short wavelengths. But the term is also used for methods that

influence Earth's emission of energy at infra-red wavelengths. A variety of SRM methods have been proposed, and some of them operate by modifying the number of submicron particles (usually called aerosols) present in the atmosphere. Aerosols have both natural and anthropogenic sources, and some SRM methods modify their number in the atmosphere in order to increase sunlight reflection, or to increase the emission of energy from Earth's surface. Changes in GHGs, aerosols and clouds change the energy budget, and the radiative fluxes in and at the boundaries of the atmosphere. Radiative drivers of climate

change are often discussed in terms of atmospheric energy flux changes that we loosely term "Effective Radiative Forcing" (ERF in $Wm^{-2}$) or just "forcing" hereafter. For brevity, we also group inadvertent forcing from aerosols, GHGs, Land Use and Land Cover changes as "GHG forcing".

One SRM strategy that has received a lot of attention in the climate community is based upon the observed cooling following strong volcanic eruptions that increase particles high in the atmosphere in a region called the stratosphere. Those increased

stratospheric aerosols reflect sunlight back to space. Methods proposing to increase the amount of light-scattering aerosols in this region of the atmosphere (by planes, rockets, balloons etc.; Budyko, 1974; Rasch et al., 2008) are therefore called "Stratospheric Aerosol Injection" (SAI).

The approach of most importance to the protocol described herein emerged from an observational and theory-based study by Slingo (1990) indicating the remarkable role that low level oceanic clouds play in Earth's energy budget, and that small changes

in those clouds could change Earth's energy budget very substantially. That study, coupled with observations of how increases in ambient aerosol concentrations within clouds can increase cloud reflectivity (Twomey, 1977), led to a suggestion by Latham (1990) that marine boundary layer clouds might be deliberately made more reflective (brightened) as a climate intervention by introducing additional aerosol particles. This strategy came to be called Marine Cloud Brightening (MCB). It was eventually recognized that not all surface-emitted particles would end up in clouds, or they may remain in the atmosphere for some time

after cloud droplets evaporate, or they could be introduced in cloud-free regions. These aerosols will also scatter additional sunlight, contributing to climate cooling, which is also known as "marine sky brightening" (MSB, Jones and Haywood, 2012; Partanen et al., 2012; Ahlm et al., 2017). The size of sea salt aerosol that would be optimal for MCB is quite a bit smaller from the size that would be optimal for MSB, and MCB isn't aimed at producing a strong direct forcing, but studies of MCB are also relevant to MSB.

The primary effect being targeted with MCB is a redistribution of cloud water from a smaller number of larger droplets to a larger number of smaller droplets, which results in greater liquid water surface area in the cloud, and therefore higher





reflectivity (known as the "Twomey effect", Twomey, 1977). However, clouds respond to this change in droplet size distribution in a number of complex ways that can change the total amount of condensed water in the cloud, referred to as the cloud liquid water path (LWP, Chen et al., 2022; Yuan et al., 2023; Khatri et al., 2023), which also affects cloud reflectivity (Wood, 2007;

Chun et al., 2023). The sign and magnitude of the LWP changes depend strongly on meteorological and background aerosol conditions as well as the magnitude of the perturbation and cloud response (Ackerman et al., 2004; Wang et al., 2010; Wood, 2007; Xue et al., 2008). The change to smaller droplets can suppress precipitation, increasing cloud LWP, but it can also increase the evaporation rate of droplets, decreasing cloud LWP. The change in droplet size can also affect how long the cloud lasts, resulting in an overall change in cloud fraction (CF). These secondary effects are often called "cloud adjustments" (Wang

et al., 2011; Jenkins et al., 2013; Chen et al., 2022) As such, the total effect of MCB must account for changes in cloud LWP and CF as well as the Twomey effect and the direct light-scattering effect of aerosols added to the atmosphere targeting cloud brightening. Importantly, many of the processes driving cloud LWP and CF changes are not fully or precisely accounted for in climate models. In particular, climate models tend to more systematically produce increases in cloud LWP and CF than is estimated from observations and from higher-resolution modeling studies that resolve and account for more complexities of

aerosol-cloud interactions (Quaas et al., 2008; Stevens and Feingold, 2009; Seinfeld et al., 2016; Malavelle et al., 2017).

In addition to accounting for these aspects specific to MCB, it is important to keep in mind that MCB, as with all SRM interventions, does not address some impacts of high $CO_2$ concentrations; most notably, SRM does not mitigate increases in ocean acidity, and the hydrological sensitivity to SRM is higher than that of GHG. Common to all SRM approaches, effective cloud brightening would, overall, cool climate, but the different SRM interventions have distinct features, different levels of

efficacy at reducing climate warming, and would affect climate benefits and risks differently. For all SRM interventions, the impact on climate changes and associated risks will depend on the specifics of how the intervention is implemented. Earth System Models (ESMs), which we also loosely call global climate models or GCMs here, allow for exploration of these responses and their dependence on implementation approach.

## 2   Background motivating the protocol

This paper introduces a protocol for computer simulations using climate models that is designed to better understand and expose possible climate consequences (risks and benefits) on decadal and longer timescales of MCB by connecting interventions in specific regions containing clouds susceptible to aerosol effects with local and far field climate responses. We also occasionally contrast MCB with SAI. Both SAI and MCB interventions produce some common responses in models of the climate system (for example, the overall cooling of the planet), but they also have very different physical characteristics and regional climate

impacts, and the level of scientific understanding differs, requiring somewhat different approaches to modeling. We will try to note some of those differences during the discussion, because they influence the protocol choices. Other SRM strategies have also been proposed and considered (see Mcnutt et al., 2015b; National Academies of Sciences, Engineering, and Medicine, 2021) but they are not discussed here. It is also important to note that the simulations specified in the protocol are an idealization



of MCB, and not a practical recipe for an implementation strategy, and they are not designed to minimize the negative climate

impacts. Those goals could be explored in a later study, or through an extension of this protocol.

## 2.1 Review of previous modeling studies of SRM impacts

This section introduces terms and concepts and to provide context for topics discussed in the rest of the paper, so is not a complete review of all previous studies. Many of the cited works can provide more detail about these topics.

Both SAI and MCB increase the reflectivity of the planet and produce a global mean negative ERF (a measure of the strength

of the cooling by a forcing agent) using aerosols, but they operate on very different space and time scales. Since SAI forcing is produced by introducing aerosols at high altitudes where their lifetime can be a year or longer, they spread (and cool) over large geographic regions. Aerosols introduced near the surface for an MCB intervention would be scavenged very rapidly (few days) so their influence covers a much smaller area. Therefore, an MCB intervention, producing an equivalent global forcing but introduced over a smaller area (usually envisioned to be 10-20% of the planet's surface, but sometimes much larger, e.g., Bala

et al. (2008); Rasch et al. (2009); Stjern et al. (2018)) would produce much stronger local ERF (cooling) than SAI. However, the cooling effect would not only be local, as winds and ocean currents can spread this cooling (Jones et al., 2009). The different characteristics of the forcing and response for MCB and SAI are potentially useful, and outcomes might be optimized through using multiple techniques in combination (Boucher et al., 2017).

Significant uncertainties remain around how any of these SRM intervention approaches would affect climate risk under

different scenarios of greenhouse gases and background aerosol concentrations. As impacts of climate change grow and become more tangible, there may be increasing pressure to consider reducing climate warming using one or more SRM approaches, but the current level of knowledge is not sufficient to detect, attribute or project with sufficient accuracy the consequences for climate risks, motivating more study of the topic.

Studies of CI have proliferated and researchers have often used differing experimental design strategies for their simula-

tions. As with other parts of the climate change research community, CI researchers recognized the advantages of developing standard scenarios (for emissions and forcing) and methodologies to compare model simulations and determine the robustness of their results, which led to formation of the Geoengineering Model Intercomparison Project (GeoMIP, Kravitz et al., 2011, 2013, 2015; Visioni et al., 2023). The GeoMIP protocols have proven very useful for identifying consistent responses across models, and also for noting features where models differed. Scientific consensus reports like the Sixth Assessment Re-

port of the Intergovernmental Panel on Climate Change (IPCC, 2021) have used studies proposed by individuals and groups, and particularly GeoMIP, to guide their conclusions about SRM. In order to maximize participation and minimize both computational cost and experimental complexity, the simulations and protocols defined for multi-model intercomparisons like GeoMIP were chosen to be quite simple, and the design setups and climate change scenarios used were not particularly realistic. The protocols were idealized by design, and they neglected, prescribed, or left unspecified many physical, chemical and

biological features that are known to be important and interact with other Earth System components, but accurate treatment of those processes is so costly (in terms of computational and human resources) that it makes sense to start simple in a common framework, and do more realistic calculations outside that framework as understanding develops. The most recent summary



of GeoMIP (Kravitz et al., 2015) outlined simulation protocols that climate modeling groups have followed to explore the climate consequences from solar dimming and from SAI and MCB climate interventions. An assessment of the strengths and weaknesses of the GeoMIP protocols is provided in Visioni et al. (2023).

There have been a few notable recent global modeling advances relevant to SRM research since those studies:

– The climate modeling community recognized the importance of using larger ensembles of simulations to better understand and characterize the role of natural variability in the Earth System and how that variability features in detecting and attributing climate change (Kay et al., 2015). These large ensembles are now sometimes used in CI research (Tilmes et al., 2018).

– In contrast to earlier SRM studies, researchers have also begun exploring the use of "controllers" to vary the location and amplitude of CI forcing to optimize it to meet specific climate objectives (MacMartin et al., 2014). One early study using a controller (Kravitz et al., 2017) developed a procedure for using SAI to manage three climate features: the globally averaged surface temperature, the difference in the warming in the two hemispheres, and the gradient in the warming from the equator to the two poles. These targets were managed by varying the amplitude of aerosol emissions in the stratosphere at four latitude-bands. These optimization procedures are designed to minimize the intervention while returning the model to a more desirable climate state. Some GeoMIP protocols have made use of a "human controller" to predict, then correct, a simulation targeting one climate metric (global averaged surface temperature).

– Modelers have begun investigating chemical (Tilmes et al. (2018); Richter et al. (2022) for SAI, and Horowitz et al. (2020) for MCB) and ecosystem impacts (Russell et al., 2012; Trisos et al., 2018).

The most ambitious study proposed for an SAI intercomparison activity (specifying a large ensemble, and a controller) to date is probably Richter et al. (2022, hereafter called ARISE-SAI). The ARISE-SAI protocol was also designed to be more policy-relevant than previous SRM protocols because it used a modern, more realistic emission scenario for anthropogenic GHG forcing and adopted a protocol intended to limit globally averaged surface temperature (rather than using SRM to address less realistic GHG scenarios) following many of the design goals discussed in MacMartin et al. (2022). The ARISE-SAI protocol started the intervention in 2030 (a more realistic but still optimistic estimate) while in the GeoMIP simulations intervention started in 2020 – which clearly is no longer feasible.While individual models performing ARISE simulations using a nominally identical controller are able to achieve and maintain multiple targets, there appear to be considerable inter-scenario differences (Wells et al., 2024), and inter-model differences (Henry et al., 2023) in the latitudinal distribution of the optimised injections strategy.

While there have been a relatively large number of climate model studies of SAI interventions, the number of studies evaluating MCB interventions is significantly smaller, and this has also resulted in a less mature and more superficial evaluation of the potential climate impacts of possible MCB implementations. Studies of MCB climate impacts have also often made differing choices about how to do the MCB intervention, by varying areal locations, extent, strength and timing of aerosol injections or forcings, and the choice of climate change scenario to which the intervention is added (e.g., Jones et al., 2009; Rasch et al., 2009; Korhonen et al., 2010; Partanen et al., 2012). Alterskjaer et al. (2013) used a variant of a GeoMIP SAI protocol



(experiment SAI G3 from Kravitz et al. (2011)) for the first MCB protocol that was designed to be used to compare multiple (in this case 3) models. A radiative perturbation from MCB (restricted to operate between 30S and 30N) was introduced to counter some of the anthropogenic forcing from an RCP4.5 scenario (Moss et al., 2008). Two models with relatively simple

treatments for clouds and aerosols constrained the cloud's radiative perturbation by explicitly overriding the number concentration of cloud drops within modeled clouds, a strategy we hereafter loosely call a "Cloud Drop Number Concentration" (CDNC) perturbation. The concentration or emission of sea salt aerosols that scatter sunlight directly was also changed, but was not required to remain consistent with the implied changes to the cloud properties. The third model employed a more complex and comprehensive treatment predicting cloud and aerosol properties, introducing the radiative perturbation by adding a source

for injected sea-salt aerosols and then letting those extra aerosols participate in all modeled physical processes. Each model was configured to produce forcing that approximately countered the GHG forcing increases between 2020 and 2070. We will generally term studies using injected Sea-Salt Aerosol emissions to produce a forcing an "iSSA perturbation" and distinguish between studies and protocols emitting iSSA optimized to be cloud nuclei, from those emitting aerosol in a size range similar to those produced naturally. The distinction is important (Connolly et al., 2014; Wood, 2021) and it is discussed more below.

More recent MCB studies have generally followed one of the paradigms described in the previous paragraph. In spite of the variations allowed for by experimental setups in these studies, some common features have emerged: first, many studies saw the expected strong local climate responses to the strong local radiative forcing connected to the regions where clouds are pervasive and susceptible; second, some far field responses (teleconnections) emerged. For example, some early MCB studies noted a tendency for MCB interventions generating strong MCB forcing in the eastern subtropical ocean basins to produce a

"La Niña"-like climate response (Rasch et al., 2009; Jones et al., 2009) even though many details of their experimental design differed. Jones et al. (2009) investigated those regional teleconnections more methodically by including and excluding specific regions from an MCB intervention to explicitly demonstrate that connection. That study and follow up work by Jones and Haywood (2012) and Hill and Ming (2012) also found a decrease in precipitation in the eastern Amazon rainforest in response to MCB forcing in the subtropical south Atlantic.

The first GeoMIP model experiments relevant to MCB (Kravitz et al., 2013, 2015) suggested studies based upon many of the studies mentioned in earlier paragraphs: 1) a change to the surface albedo over all ocean regions; 2) an increase in CDNC by 50% ; 3) an increase in natural sea salt emissions; and 4) the Alterskjaer et al. (2013) multi-model protocol summarized in a few sentences above. While useful for a first estimate of MCB potential effects on climate, runs with highly simplified representation of MCB (e.g. changing the surface albedo over all oceans, or all marine clouds equatorward of 30°) clearly will

produce a very different climate response than realistic implementations of MCB.

    Other aspects of the CDNC and iSSA perturbation runs were also unrealistic. For example, simulating MCB by increasing the natural sea salt emissions introduces aerosol particles that much larger than those considered optimal for MCB (e.g., Wood, 2021). Expressing "MCB efficiency" as being a measure of the radiative forcing per mass of injected salt aerosol, these larger aerosols are not ideal for brightening clouds, but they are more efficient at scattering sunlight directly – leading to different

conclusions about both the mass of sea salt aerosol that would need to be emitted to achieve a given forcing and the relative roles of clear sky MSB versus forcing through cloud brightening that would be produced by MCB.





Our modeling protocol builds on these earlier studies, moving towards more realistic representations of MCB that can be simulated across multiple models. We seek to bridge the gap between past regional MCB studies that made different implementation choices impeding clear model comparisons, and the GeoMIP simulations that deployed less realistic uniform interventions over extremely large marine regions. By performing an inter-model comparison of the MCB effect for consistently defined regional interventions, we aim to clarify the key points of agreement and uncertainty in the climate response to more plausible MCB deployments.

## 2.2 Limitations and Strengths of global models for MCB evaluation

Models used for evaluating climate change and MCB intervention have been found to be very sensitive to the manner in which aerosols, clouds, and their interactions are treated. The physical and chemical processes involving aerosols and clouds are among the most challenging, complex, and difficult to represent in atmospheric models (Stevens and Feingold, 2009; Carslaw, 2022). Approximations and simplifications required for treatment of these processes in climate models are responsible for many variations in model behavior when those tools are exposed to present day aerosol perturbations (Malavelle et al., 2017), and produce a lot of uncertainty when used to study historical changes in climate and to project future changes (Masson-Delmotte and et al, 2021). Parameterizations of aerosols and the responses of clouds to aerosols in climate models show quite different responses across models (Ghan et al., 2016; Malavelle et al., 2017). Assessments of the performance of global models from large scale emissions from effusive volcanic eruptions suggests that they are able to represent the impact on cloud effective radius with reasonable fidelity (e.g., Malavelle et al., 2017). However the observed impacts on the liquid water path and in particular the cloud fraction that appears to be a key and often overlooked mechanism in climate models (e.g., Chen et al., 2022, 2024) diverge widely between models, and these responses also differ from those seen in observations and simulated with detailed Large Eddy Simulation (LES) models that can explicitly treat most of the relevant physics. LES models (Wang and Feingold, 2009; Wang et al., 2011; Possner et al., 2018) are regarded as providing a much more reliable portrayal of the aerosol-cloud interactions and local cloud feedbacks that are important to the study of climate change and of MCB. However, they are not useful for exploring questions on large time and space scales because it is far too expensive computationally to run simulations longer than a few days over domains large enough to capture regional and global-scale climate and Earth System responses.

Climate models are however, very powerful tools for examining responses from forcing agents (for example to MCB interventions) through energy and water budget changes, or via circulation feature changes (temperature, winds, ocean currents, precipitation, soil moisture, etc.) through interactions and feedbacks occurring between climate system components (atmosphere, ocean, land, ecosystems). So it is important to use both small scale process models, and climate models for exploring different science questions associated with MCB intervention.

## 2.3 Considerations influencing the Protocol Design

The issues discussed above guide some aspects of our specification of a protocol for exploring climate responses to MCB. While we do not assume that climate model results can necessarily provide quantitative information about how much a CDNC



perturbation to clouds or SSA emissions will be required to produce a specific MCB forcing or a given circulation response, we are interested in the range of these responses, and in identifying the implications of the required perturbations to the feasibility of an MCB intervention. We have designed our global modeling protocol to focus on quantifying the climate responses to a given forcing in particular region(s) rather than on simulating aerosol-cloud interactions with high fidelity. We use the simulations to provide information about the range of uncertainty across models in achieving these climate responses, and to document the model's cloud responses to a perturbation. As described below, global models often respond very differently to perturbations that are intended to be identical, such as to a specific amplitude of perturbation in CDNC or sea-salt emissions. The magnitude of changes in these perturbations needed to achieve approximately the same MCB forcing differs, and the forcing is often produced by differing cloud responses, and by differing contributions from the aerosols in cloud free regions. However, as also will be shown below, there is a strong level of consistency in the response to a given regional forcing (regardless of how that forcing was achieved) among the models.

**Table 1.** List of questions this MCB protocol for climate models is targeting

| | |
|---|---|
| Q1 | Do recent climate models indicate it is feasible to produce MCB forcing that would counter a substantial fraction of GHG forcing? |
| Q2 | Which regions do models suggest produce strong forcing, and/or strong cooling? Which cloud regimes? Are there characteristic, robust responses to forcing in specific regions? |
| Q3 | How many seeding areas are needed to achieve substantial forcing? |
| Q4 | How is the MCB forcing achieved (Twomey effect, cloud adjustments, direct forcing)? What are the local signatures? Are they consistent across models? With relevant observations and/or with cloud-resolving (LES) simulations? |
| Q5 | Do feedbacks make particular regions more effective than others in producing a response (e.g., are there 'hotspots' that amplify or damp the climate response to an MCB intervention)? |
| Q6 | Which teleconnections would be affected by an MCB intervention? Are there common and robust signatures across multiple models that an MCB intervention in a specific region will trigger a teleconnection (far field) response? |
| Q7 | Are the impacts of an MCB intervention in multiple regions additive (and linear)? |
| Q8 | Are there trade-offs with interventions in specific regions, or compensations when forcing in multiple regions? |

As previously mentioned, there have only been a few multi-model intercomparison activities designed to understand climate responses to regional MCB and MSB forcing, and there are many questions that have emerged from previous research. This study proposes extensions intended to provide a more systematic and internally consistent evaluation of the climate impacts of different feasible MCB implementations and a better understanding of why the responses in global models differ. The protocol is designed to help address some of the questions appearing in Table 1, which are a more granular sub-set of the research questions raised by Diamond et al. (2022).

Question Q1 is intended to move MCB investigations away from using the RCP 8.5 (or its equivalent SSP) scenario that has often been used in past SRM studies; they do not appear to be very realistic or policy relevant. Earlier MCB studies like those of (Rasch et al., 2009) or GeoMIP (Kravitz et al., 2013, 2015) that introduce the MCB intervention over very large oceanic



regions are very broad-brush and don't provide much insight into which regions matter or what their impacts are. No studies have really looked at the relative susceptibility of some regions compared to others, and the reasons for that susceptibility. Question Q2 is designed to quantify this, and identify whether there are common signatures in cloud responses across models, or common signatures (in pattern, and amplitude) of the circulation response to an MCB ERF imposed in specific regions. An understanding of the forcing response to a iSSA perturbation is also a necessary first step if a multi-region automated

controller is going to be considered in future studies. And although some previous studies (e.g., (Jones et al., 2009; Hill and Ming, 2012) have sometimes identified common signatures in multiple models the areal extent and location has still differed pretty substantially (factors of 2 or more) so it is difficult to compare ERF and circulation responses quantitatively. Question Q3 returns attention to whether MCB might be feasible in order to counter some impacts of GHG forcing, and how large the ocean area must be to produce such a cooling. Question Q4 is intended to reveal whether the forcing is achieved by similar

mechanisms in different models, and to provide enough information about the cloud changes that some comparisons can be made with observations or detailed cloud models. It is also possible that certain regions are more sensitive to an intervention than others (addressed in question Q5). We show below that models often behave differently and that use of a common carefully specified protocol helps to expose those differences. Question Q6 focuses on teleconnections. We show below that a superficial comparison of three models confirms the previously identified "La-Niña"-like SST response, but also show that the amplitude

of cooling in the eastern pacific is quite different in those models. Boucher et al. (2017) examined the radiative impacts of combined use of MCB and SAI in fixed SST simulations, and concluded that forcing and rapid atmospheric adjustments from the two SRM methods were additive and concluded by noting the need for coupled simulations to assess the implications for additivity of climate responses. We extend this approach in question Q7 by evaluating the additivity of radiative forcing and climate responses to MCB forcing in multiple regions. Lastly question Q8 expresses our interest in consequences of CI in

different regions and whether interventions in combinations of regions might amplify signatures in specific regions or be used to compensate for specific features.

**Table 2.** Stages: see text for more discussion

| Stage # | Setup | Goal |
|---------|-------|------|
| Stage-1 | Short control and regionally-focused perturbation simulations with fixed SSTs. | Document response in forcing and clouds to perturbations for calibration |
| Stage-2 | Coupled simulations with MCB implementation set to achieve a given forcing in individual regions, to identify teleconnection patterns, followed by (one or more) simulations with forcing in concurrent regions | Document coupled responses in circulation features to perturbations (feedbacks, teleconnections) and establish whether there are important nonlinear interactions. |
| Stage-3 | Coupled simulations designed to achieve specific climate objectives. The coupled simulation can, but may not be required to, use optimization algorithms (e.g. controllers). | These simulations would be intended to be policy relevant and useful for risk/benefit assessments |



## 3    The Protocol

Our protocol is intended to investigate MCB in 3 major stages (see Table 2). The first stage establishes characteristics of the radiative forcing produced by the intervention. This is particularly important for MCB/MSB interventions because of

uncertainties in clouds' physical responses to aerosol perturbations, and the particular difficulties that global models have in capturing these interactions, as discussed in section 2.3. Climate model simulations with fixed Sea Surface Temperatures (SSTs) are used to establish the radiative flux changes caused by CDNC and iSSA perturbations to: 1) establish whether current climate models are capable of producing a strong global forcing response to feasible perturbations; 2) assess whether cloud responses differ from region to region; 3) identify whether the parameterized aerosol and cloud responses appear physically reasonable;

and 4) evaluate the linearity of the responses to perturbations (e.g, whether forcing scales linearly with the amplitude of the perturbation and whether a forced response in one region is sensitive to forcing taking place in other regions).

The second stage uses coupled GCM simulations with idealized perturbations (in which the atmosphere, ocean and sea-ice can fully interact) to expose changes in circulation features. Perturbations are introduced to identify: 1) how climate feedbacks operate on the forcings; 2) whether teleconnections exist producing far field responses; and 3) whether the coupling of Earth

System components introduces nonlinear interactions or interactions between regions.

The first and second stage simulations are intended to provide data required to design a planned third stage of "scenario" simulations, in which MCB is adjusted over time to maintain selected climate targets (similar to ARISE-SAI, but applied to MCB interventions). Such simulations defined for stage three of the project should be designed to be policy relevant for feasible interventions and useful for risk/benefit assessments. This set of simulations could include use of a "controller", where MCB

implementation locations and amount are adjusted by the controller in a way that targets specific climate metrics, such as has been done for SAI as described above.

We do not prescribe specifics of the stage 3 simulations here, as the design of the MCB scenarios will depend strongly on the outcomes of the first and second stage simulations. Analysis of the simulations from stage 1 and 2 (particularly if a larger group of models contribute simulations) should be useful in guiding the stage 3 design. Development of a controller in particular will

rely on better understanding of how MCB implementation in different regions affects different climate metrics, and decisions around what climate metrics might be best targeted using MCB.

### 3.1    Configurations, Setup, Experiments:

While some SRM studies have used "Slab Ocean Models" that account for an ocean thermodynamic response, but prescribe ocean dynamics and heat transport, we favor use of full ocean models. Slab models provide an inexpensive first order estimate

of ocean temperature change, but don't allow realistic dynamic ocean responses that may be important (e.g. Meridional Overturning Circulation (MOC) and El-Niño Southern Oscillation (ENSO) responses) that are sensitive to ocean atmosphere heat and salinity fluxes. Modelers should include dynamic ocean and sea-ice components that have been coupled with a baseline simulation for 50 years or longer prior to the start of simulations (as described in Table 4) so that the mean state of the upper ocean, sea-ice and atmosphere are not drifting strongly due to the choice of ocean/sea-ice initial conditions. Optionally, it might



also be interesting to perform slab-ocean simulations because some teleconnection-enabling processes like wind-evaporation-SST feedbacks will still operate in that configuration (Xie and Philander, 1994; Mahajan et al., 2009), but we encourage modelers to begin with fully-coupled configurations.

Our protocol is similar in spirit to Jones et al. (2009, hereafter JHB2009) and Jones and Haywood (2012, hereafter JH2012): we selectively introduce forcings in individual regions, and then introduce forcing in multiple regions concurrently. The proto-
col differs from JHB2009 in some ways: rather than identifying irregular regions each region is regular and occupies approximately 4% of the global ocean area (see Table 3 and Figure 1), with the exception of the Arctic region, NO. The NO region has an areal extent less than half that of the other regions, and since the Arctic is ice covered for substantial part of the year the iSSA emissions are reduced substantially. Each region has persistent marine boundary layer clouds known to be susceptible to brightening from aerosols. The regions' regularity makes it easier to specify across different models, and allows a definition
that does not depend on characteristics of the model grid or parameterization details. The larger area used in this protocol also allows cloud deck locations to differ somewhat from model to model. Aerosol perturbations are introduced only in model cells containing at least 90% open ocean (that is, models should not perturb columns containing significant land or sea-ice). The protocol also assumes that the amplitude of the perturbation will be chosen through a calibration procedure. The amplitude is controlled by specifying a single "scaling" parameter that controls the CDNC value (drops per unit volume), or the iSSA
emission (kg per unit area of particles of specific size). For stages 1 and 2, a common scaling value is used in every oceanic region involved in the intervention. So, for example for an iSSA intervention involving regions R1, R2, and R3, the same sea salt mass per unit area is injected into each of the three regions, to avoid too many permutations of forcing and to expose differing sensitivities in each region. Future studies might use differing perturbation amplitudes region by region to optimize a particular outcome, but that is beyond the scope of the present protocol.

Our strong preference is that the perturbation be introduced using a sea spray source similar to that proposed by Latham et al. (2013), that is capable of affecting clouds via emissions of 50-100 nm dry diameter soluble aerosol. Some models may find it difficult to specify this type of sea salt emission source, so it is also acceptable (but deprecated) to directly perturb CDNC in clouds in the specified regions.

We anticipate that perturbations of the same amplitude (in iSSA emissions or CDNC) introduced in two different models will
produce different forcing. Our primary interest is in evaluating the climate responses to total (direct+indirect) forcing of similar amplitudes, so the protocol allows the perturbation amplitude across models to differ. A variable controlling the amplitude of the perturbation within all regions is chosen to produce a similar globally averaged forcing when applied concurrently to the NEP, SEP, and SEA regions identified in figure 1 and table 3. This produces forcing that is largely distributed over the stratus/stratocumulus regions off California, Chile/Peru and Namibia where stratus and trade cumulus clouds are commonly
observed. These regions have often been focused on in past studies, e.g., JBH2009, JH2012, Rasch et al. (2009) and Hill and Ming (2012). Our protocol formalizes the choice of regions, and the amplitude of the forcing imposed in the three regions so that the global forcing is approximately the same when each model is forced in those three regions.

For consistency, we use the same size perturbation strength (in iSSA or CDNC) in each region where MCB is implemented for the regions described in table 3, to assess model responses to forcing in those regions. We anticipate that both forcing and




responses will differ from one region to another because background aerosol amounts and meteorological regimes differ from one region to another.

The mid-latitude regions in the South and North Pacific Ocean (SP and NP) and in the northern polar ocean region (NO) contain many clouds driven very strongly by mid-latitude dynamical features and so have not been a focus of MCB studies to date (although they were used in Rasch et al. (2009) and Haywood et al. (2023, hereafter HEA2023) among others). These regions also have a high fraction of marine low clouds that may be viable targets for MCB, as indicated by recent observational studies showing significant cloud brightening with aerosol perturbations (Mace et al., 2023; Chen et al., 2022; Murray-Watson and Gryspeerdt, 2022). As such, we also include these regions in our protocol. The SP and NP regions are included for some of the simulations to understand what leverage one might achieve with forcing further from the equator.

**Table 3.** The regions to be evaluated that have persistent low clouds susceptible to aerosols. Each region occupies about 4% of the global ocean except NO (2%). Interventions are introduced only over columns containing open-ocean fractions exceeding 90%. See also Figure 1 and text for more detail.

| Region | Region (abbrev) | Latitude Range | Longitude Range |
|---|---|---|---|
| R1 | NE Pacific (NEP) | 0-30°N | 110°W-150°W |
| R2 | SE Pacific (SEP) | 0-30°S | 70°W-110°W |
| R3 | SE Atlantic (SEA) | 0-30°S | 15°E-25°W |
| R4 | N Pacific (NP) | 30°N-50°N | 170°E-120°W |
| R5 | S Pacific (SP) | 30°S-50°S | 90°W-170°W |
| R6 | Northern Oceans (NO) | 60-90°N | 0°W-360°W |

## 3.2 Calibration Procedure

This protocol requests that the MCB perturbation be calibrated to produce a target global annually averaged effective radiative forcing (ERF) from the combination of cloud and (for MCB introduced through an iSSA perturbation) direct aerosol radiative forcing of between -1.5 and -2 W m$^{-2}$ in the fixed SST simulations with the intervention in the regions NEP+SEP+SEA concurrently, which constitutes the first stage of our protocol (see Table 2). This targeted total global forcing magnitude is about half the amplitude (and of opposite sign) of that associated with a $CO_2$ doubling.

The nominal MCB perturbation is achieved by starting with a series of short (1-5 year) "fixed Sea Surface Temperature (SST)" simulations to identify the required amplitude of the perturbation (with respect to a control simulation using prescribed AMIP or climatological SSTs for the decade around year 2000) needed to reach the target global forcing. For models using specified CDNCs to achieve the forcing we suggest that CDNC perturbations be introduced by starting with a CDNC perturbation of 375 cm$^{-3}$ in each region, then scaling the perturbation up or down using short CDNC simulations until the forcing produced from the 3 regions reaches a target of about -1.8 W m$^{-2}$. Models using a sea-spray source should start by setting injected sea-salt emissions to about 50 Tg y$^{-1}$ (NaCl, ignoring contribution from other salts and organics) in each region (150 Tg y$^{-1}$ for the sum of the NEP, SEP, and SEA), then scaling the emission rate up or down to achieve that same target.





Because the protocol allows a lot of flexibility in how the MCB forcing is introduced, there can be substantial differences between models in the resulting partitioning between direct (clear-sky) and indirect (cloudy-sky) forcing within each region.

Temporal variations in forcing may also differ because of variations in the seasonal cycle of the clouds and climate. The MCB perturbations are introduced so the cloud responses are primarily local to and possibly somewhat downstream when introduced as an iSSA perturbation. For the rest of the paper the perturbation nominal value is that which produces (either by specifying the iSSA emissions rate (kg m$^{-2}$ s$^{-1}$), or CDNC value (# cm$^{-3}$)) a total forcing (ERF) of about -1.8 W m$^{-2}$ from concurrent interventions in the NEP, SEP and SEA regions in a simulation with prescribed SSTs. This value is a pretty rough estimate of

the forcing, because it doesn't take into account natural interannual variability of climate variables, nor variations in SSTs, the background concentration of aerosols, or other factors/feedbacks that might cause variations in ERF, but it is sufficient to make sure that models are producing about the same ERF, and it allow us to establish the nonlinearity in responses to perturbation amplitude (examples are shown in Section 3.3). We suggest that the same perturbation value (e.g., iSSA emissions rate per square meter, or in-cloud CDNC value) be used for the simulations involving perturbations in other regions, so when more

regions are included in an intervention the global forcing with perturbations using a nominal value could be quite different. In the models considered here, a larger areal extent of the perturbation will produce stronger (more negative) forcing (than -1.8 W m$^{-2}$) since the introduced perturbation produces negative forcing in all regions.

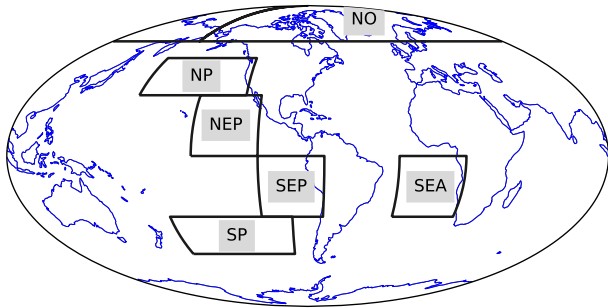

**Figure 1.** The MCB regions. Also see Table 3.

### 3.3 An outline of the protocol's experiments and a few illustrative results

This section provides text motivating the experiments, provides caveats, and shows some illustrative examples of conclusions

that can be gleaned from those simulations. A brief description of the models used in the examples, and details of the implementation of the MCB perturbation for each model is provided in Appendix A1. Table 4 summarizes the simulations (experiments) needed to complete Stages 1 and 2 of the protocol. Some notes describing requested model output are provided in appendix A3.





**Table 4.** Table of experiments: Y2000AF means the simulation is run with Anthropogenic Forcing and SSTs for the decade centered on year 2000. SSP-baseline is ideally the SSP2-4.5 scenario.

| Simulation ID | Model Configuration | Run length | Rationale |
|---|---|---|---|
| Stage 1 | | | |
| E1 | Fixed SST, Y2000AF | 15 yr simulation | control runs |
| E2-cal | Fixed SST, Y2000AF+MCB | multiple short (< 5 year) simulations | calibration to establish the model's nominal perturbation value when MCB is introduced in NEP+SEP+SEA (see section 3.2) |
| E2-NEP | Fixed SST, Y2000AF+MCB | 15 yr simulation | ERF in NEP |
| E2-SEP | Fixed SST, Y2000AF+MCB | 15 yr simulation | ERF in SEP |
| E2-SEA | Fixed SST, Y2000AF+MCB | 15 yr simulation | ERF in SEA |
| E2-NP | Fixed SST, Y2000AF+MCB | 15 yr simulation | ERF in NP |
| E2-SP | Fixed SST, Y2000AF+MCB | 15 yr simulation | ERF in SP |
| E2-NO | Fixed SST, Y2000AF+MCB | 15 yr simulation | ERF in NO |
| E2-NEP+SEP+SEA | Fixed SST, Y2000AF+MCB | 15 yr simulation | ERF in NEP+SEP+SEA |
| Stage 2 | | | |
| E3 | Coupled, SSP245 | 30 to 80 yr simulation | control runs |
| E4-NEP | Coupled, SSP245+MCB | 30 to 80 yr simulation | ERF in NEP |
| E4-SEP | Coupled, SSP245+MCB | 30 to 80 yr simulation | ERF in SEP |
| E4-SEA | Coupled, SSP245+MCB | 30 to 80 yr simulation | ERF in SEA |
| E4-NP | Coupled, SSP245+MCB | 30 to 80 yr simulation | ERF in NP |
| E4-SP | Coupled, SSP245+MCB | 30 to 80 yr simulation | ERF in SP |
| E4-NO | Coupled, SSP245+MCB | 30 to 80 yr simulation | ERF in NO |
| E4-NEP+SEP+SEA | Coupled, SSP245+MCB | 30 to 80 yr simulation | ERF in NEP+SEP+SEA |
| E4-NEP+SEP+NP+SP | Coupled, SSP245+MCB | 30 to 80 yr simulation | ERF in NEP+SEP+NP+SP |

### 3.3.1 Simulations using prescribed Sea Surface Temperatures

Experiment E1 is used to define the Fixed SST control simulations. The E2-cal experiments are to establish the range of forcing (in the absence of feedbacks) that can be produced by a particular perturbation strategy and to provide estimates of the nominal perturbation value needed to produce substantial cooling for a specific perturbation strategy. These simulations also expose differences in the modeled aerosol-cloud-radiation interactions (e.g., changes in cloud drop size, LWP, or CF) under the differing perturbation strategies, but not circulation changes. Once the nominal perturbation value is known, the other E2 385 simulations are then used to produce quantitative estimates of the ERF produced from iSSA or CDNC perturbations in a range of regions, and to establish whether the ERF operates in an approximately additive manner (when perturbations are introduced concurrently with fixed SSTs). The perturbation amplitude is expected to be approximately the same in each region, but the



radiative (e.g. due to differing baseline aerosol concentrations or CDNC) and circulation responses may differ from region to region , and the total response will depend on the choice of regions, and number of regions involved in the intervention.

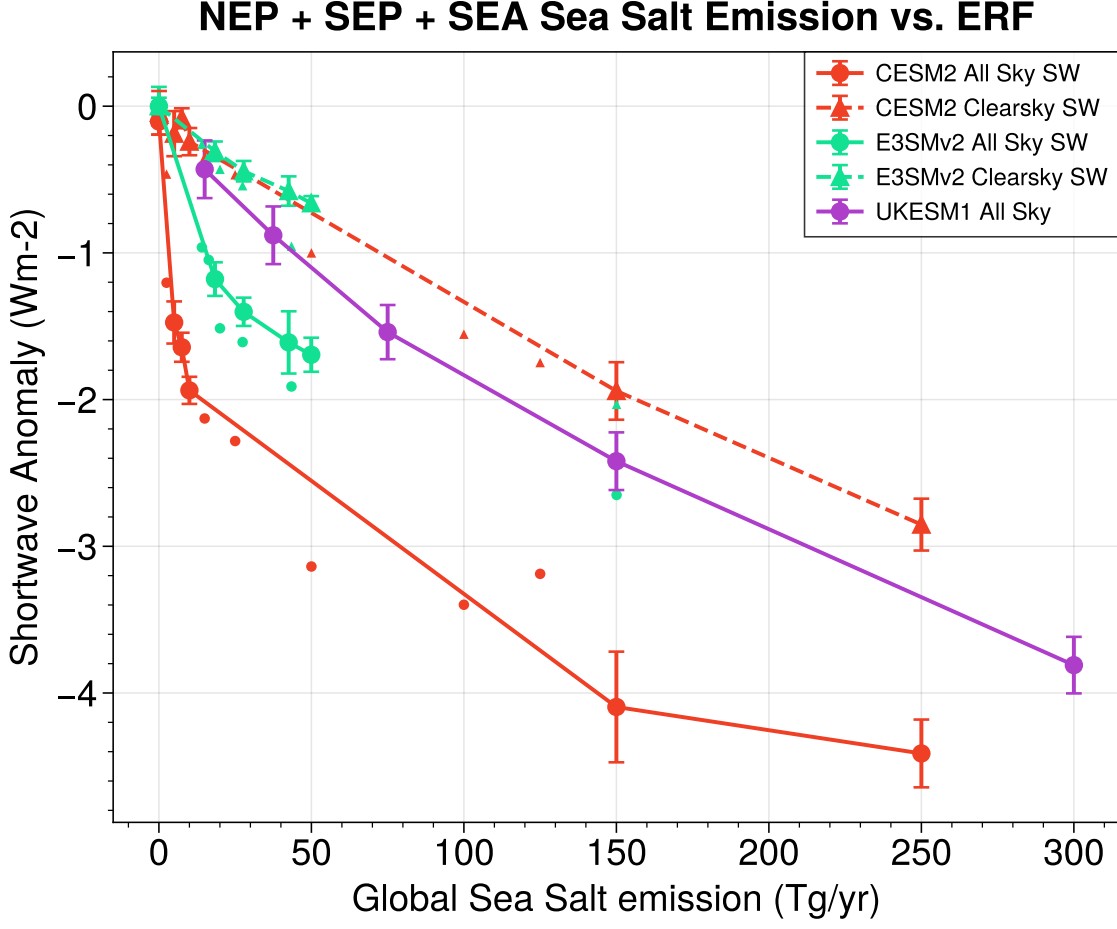

**Figure 2.** Shortwave ERF estimated from Fixed SST simulations for R1+R2+R3 emissions simultaneously (CESM2 and E3SMv2) or computed as the sum of individual regional experiments (SYN) (UKESM1). Dashed lines for CESM2 and E3SMv2 show the clearsky component of the forcing. Small dots show all-sky ERF computed from shorter 1-year testing simulations.

Figure 2 provides an example of the substantial differences between the E3SMv2, CESM2, and UKESM1 models in the cloud responses to a range of iSSA perturbations introduced within the NEP, SEP, and SEA regions. The CESM2 model required iSSA emissions of approximately 7.5 Tg/yr to achieve a forcing of about -1.8 Wm$^{-2}$ and the forcing is almost entirely from the cloud response to the iSSA emissions. The E3SMv2 and UKESM1 models require much stronger emissions (up to a factor of 10 increase), and a substantial fraction of the forcing is due to direct sunlight reflection by the added aerosols. Table 5 shows estimates of the required CDNC perturbation and iSSA Emissions needed to reach the target forcing for those models. E3SMv2 is not able to achieve that forcing until CDNC is increased to 2000 cm$^{-3}$. If the E3SMv2 model is correctly



representing the change in cloud albedo produced by this aerosol perturbation it appears unlikely that the target forcing could be achieved through feasible iSSA emissions in these 3 regions only, since such CDNCs are only present in exceedingly polluted environments (e.g. Quan et al., 2011). Ramanathan et al. (2001, Fig.5) summarizes different observational datasets for marine low clouds that suggests it is rare to find cases with CDNC >500 cm$^{-3}$ even when aerosol concentrations rise into the low thousands.

The E3SM configurations using iSSA scenarios can actually achieve the target forcing with substantially lower CDNC changes than the simulations where CDNC is perturbed directly, for a few different reasons. First, overwriting the calculated CDNC with a prescribed value can lead to restrictive and internally inconsistent states within a model in terms of cloud drop size distribution properties that govern cloud microphysical and radiative processes. Most models will attempt to "iron out" those inconsistencies as each process parameterization is invoked and particular cloud properties are subsequently used, but the adjustments are imperfect, particularly when concentrations are specified that are well outside the range of values the model would produce for a particular (meteorological and aerosol) regime. These inconsistencies are not present (or appear much smaller) when the perturbation is introduced as an aerosol source. Secondly, the sea salt aerosol can spread via atmospheric transport and turbulence and occupy a somewhat larger region and depth than when the perturbation is induced as a CDNC perturbation. Those slightly more extensive cloud changes also contribute to the forcing. Thirdly, the aerosol direct effect can be very important in achieving a target forcing, and the CDNC perturbation strategy doesn't capture this forcing component.

Question Q7 of Table 1 considers the additivity of cloud and circulation responses to perturbations when they are introduced separately or concurrently in different regions. An estimate of the forcing introduced by a perturbation in a set of regions can be made by adding the changes found in simulations with individual region perturbations (which we term hereafter a "Synthetic" estimate of the model quantity) and comparing that estimate to a simulation where the regions are perturbed concurrently. Figure 3 shows the global averaged SYN and Concurrent forcing estimates for iSSA perturbations in CESM2 and E3SMv2. There are no statistically significant differences between the two global averages, as the bootstrap resampled distributions of the Concurrent minus SYN global means overlap zero for both models (see values shown in Fig. 3e,f). Spatial patterns are also similar. Thus, we see little evidence for significant non-additivity in the ERF with the inclusion of iSSA perturbations in different regions.

Figure 4 shows the globally averaged forcing response of the three models to iSSA introduced independently in each of the six regions and for combinations of regions (both concurrent and synthetic estimates). Estimates are provided for both NEP+SEP+SEA regions as well as the NEP+SEP+NP+SP regions that are part of the stage 2 simulations. As with the CDNC perturbations displayed in figure 2, the forcing was found to scale nonlinearly with the perturbation amplitude (not shown) and the nonlinearity is most evident for low emissions where the cloud response dominates. The forced response varies by region. In UKESM1 the larger responses are found in the southern hemisphere (SP, SEA, and SEP), with a smaller response in the NEP and NP regions; the NO region occupies a much smaller areal extent, and since the emissions are also proportional to the region of ice-free ocean, the smaller response there isn't surprising. The comparison of concurrent and synthetic estimates suggests the synthetic method is reasonable when circulations responses are suppressed by prescribing the SSTs.





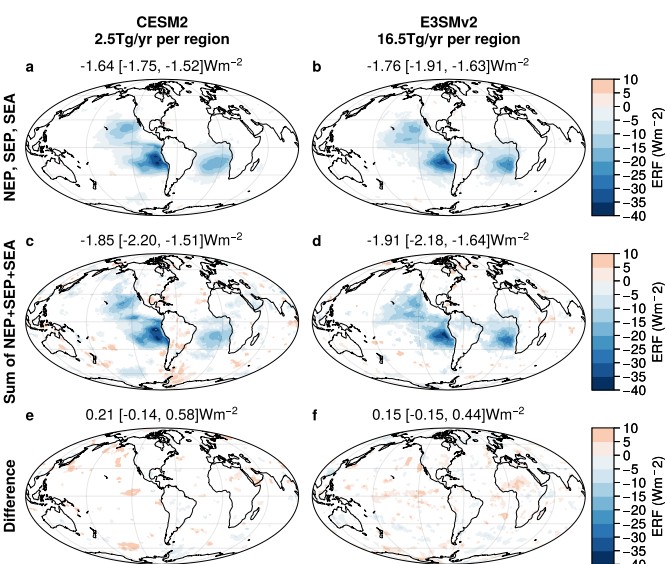

**Figure 3.** Top-of-the-atmosphere radiative flux anomalies from fixed SST simulations. iSSA emissions in CESM2 at 2.5Tg y$^{-1}$ per region (left column) and E3SMv2 at 16.5Tg y$^{-1}$ per region (right column) in R1,R2,R3 simultaneously (top row) versus the sum of ERF fields when aerosol is emitted in each region separately (the synthetic estimate; middle row) and the difference between the two (bottom row). Non-significant grid points by the t-statistic and false detection rate test are masked with white. The title of each panel displays the global mean ERF and the 5-95 percentile range of the bootstrap resampled mean.

**Table 5.** Nominal CDNC values or iSSA emission rates and associated effective radiative forcing values for NEP,SEP,SEA Fixed SST simulations.

| Model | Strategy | Perturbation amplitude | Estimated forcing |
|-------|----------|------------------------|-------------------|
| CESM | CDNC | 600cm$^{-3}$ | -1.8 Wm$^{-2}$ |
| CESM | iSSA | 2.5Tg/yr/region | -1.8 Wm$^{-2}$ |
| E3SM | CDNC | 2000cm$^{-3}$ | -1.8 Wm$^{-2}$ |
| E3SM | iSSA | 16Tg/yr/region | -1.8 Wm$^{-2}$ |
| UKESM | iSSA | 25Tg/yr/region | -1.7 Wm$^{-2}$ |

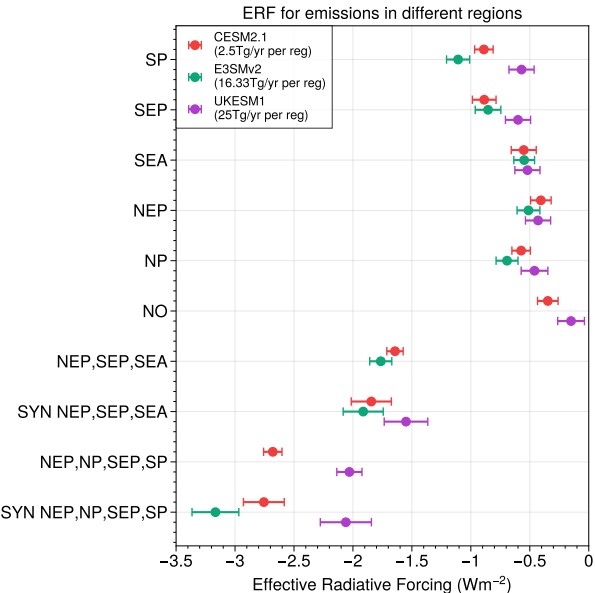

**Figure 4.** Effective Radiative Forcing (ERF) computed from the top of atmosphere downward shortwave and longwave radiative flux anomalies in fixed SST simulations for sea salt emissions in different regions and models. Error bars show the two standard error range. The nominal single region emission rate shown in the legend (CESM2 = 2.5Tg yr$^{-1}$, E3SMv2 = 16.5Tg yr$^{-1}$, UKESM1 = 25Tg yr$^{-1}$) is one-third the emission rate that causes ERF = -1.8Wm$^{-2}$ when emissions are imposed in NEP,SEP,SEA simultaneously. Thus, NEP,SEP,SEA emissions are 3x the nominal values and NEP,SEP,NP,SP emissions are 4x the nominal value. Concurrent NEP,SEP,SEA is not available for UKESM and NO is not available for E3SMv2.

Table 5 and figure 5 show how strongly the atmospheric response to sea spray emissions differs in three models calibrated to produce approximately the same global forcing in the three regions most frequently considered for MCB (NEP, SEP and SEA). The UKESM1 (SYN estimate) requires approximately 10 times higher iSSA perturbation than CESM2 to get the same forcing. The emissions required to achieve a target forcing in the UKESM1 and E3SMv2 models are more similar, and the models' cloud responses are also closer. Because the CESM2 boundary layer clouds are so susceptible to aerosols the total forcing has been achieved with a smaller change in clear-sky radiative fluxes in that model (-0.3Wm$^{-2}$) than the other two models, which require much larger iSSA emissions, such that 40-60% of the total forcing is through the direct radiative forcing by the aerosols. Much of the cloud response in CESM2 occurs through changes in cloud cover, while the other two models have a relatively muted response in this field. The CESM2 also produces a much larger response through increases in the cloud Liquid Water Path (LWP).

### 3.3.2 Coupled Simulations

The impact of CI in a coupled climate model framework can be assessed using a variety of strategies. One straightforward method used here compares simulations with (fixed or varying) GHG forcing and (fixed or varying) CI forcing to baseline



**Figure 5.** Forcings and Cloud Responses produced by iSSA perturbations in fixed-SST simulations designed to produce similar global, annual average forcing in 3 models. Changes in shortwave (allsky and clearsky) fluxes, Cloud Cover, and Liquid Water Path (LWP) are shown by rows, respectively, for the UKESM, E3SM, and CESM models (left to right by column respectively). A synthetic estimate is used for UKESM, and the Cloud Cover field estimated as the maximum Cloud cover at pressures > 850hPa.





simulations without an SRM intervention. Other approaches are also useful, for example by comparing model runs with CI to
a pre-industrial period or period prior to the beginning of the intervention. Modelers have also often compared the CI+GHG
simulations to a simulations with weaker GHG forcing (Haywood et al., 2023).

Our simulations show the MCB global mean cooling effect does not significantly change when comparing the MCB effect
under early- versus mid-21$^{st}$ century SSP2-4.5 warming in CESM2 and the difference in response is modest in E3SMv2. The
MCB cooling pattern is also very similar between these periods, though statistically significant differences are apparent in the
North Atlantic for CESM2 (appendix Fig A5 e,f). Based on this, the details of the GHG forcing trajectory and the choice of
baseline appear to be secondary in answering some of the question in Table 1. The CESM2 simulations do show some non-
linearities between the GHG and MCB in the northeast Pacific (see also Wan et al., 2023) so the baseline scenario may influence
the response in some models. These issues are discussed more in text surrounding figures 11, 10 and section 4. Our protocol
encourages use of the SSP2-4.5 (Riahi et al., 2017) or a similar scenario (e.g. RCP4.5) with coupled ocean, sea-ice and land
model components. The critical features to participate in this assessment protocol are to use MCB forcings that are of similar
amplitudes in common regions. Our experimental design and evaluation strategy is a variant of that used in GeoMIP studies to
date, extending it by allowing either an SSP scenario to be used as a control (the experiment listed as "E3" in Table 4), and the
perturbation simulation (listed as experiments "E4" plus a qualifier in Table 4) that includes the control's GHG forcing as well
as forcing from the MCB intervention. Our preference for the E4 experiments is to impose a fixed MCB forcing superimposed
upon an SSP2-4.5 baseline simulation similar to the "G7" Cirrus Thinning protocol, first described in Kravitz et al. (2015), but
we also allow and discuss some simulations using time-varying emissions intended to counteract a time-varying target forcing,
as in the "G6" experiments described in Kravitz et al. (2015) or HEA2023. We note that the pattern of response in some climate
variables is broadly similar for simulations using CDNC and iSSA perturbations in CESM2 and E3SMv2 if the location and
strength of the radiative forcing is similar (e.g., temperature - Fig. A3; precipitation - Fig. A4), but more study would be useful
to assess whether the CDNC perturbation method is a robust framework for evaluating MCB impacts.

Figure 6 shows the time series of global, annual average surface temperature (TS) in CESM2 and E3SMv2 for a series of
control and perturbation simulations using an SSP2-4.5 scenario for the baseline and control, with perturbations capable of
producing an additional approximately -1.8 Wm$^{-2}$ MCB intervention when introduced concurrently in the NEP, SEP, and
SEA regions. The simulations' global average TS change appears quite insensitive to whether MCB is introduced as a CDNC
or iSSA perturbation. The differences between the two perturbation strategies are no larger than the variability revealed from
the 3 members of the control ensemble of CESM2, at least for this measure of climate change. The CESM2 model surface
temperature response to MCB is somewhat larger than in the E3SMv2 model for a similar ERF, and the adjustment occurs more
rapidly, presumably because of different ocean model characteristics. The globally-averaged TS response to the instantaneous
application of the MCB forcing appears quite quickly (within a couple of years), and the increase in temperature from the
growing GHG forcing reappears after 20 years or so.

Figure 7 shows the precipitation response in UKESM1, E3SMv2 and CESM2 for interventions introduced in individual re-
gions through iSSA perturbations for short, coupled simulations (averages of years 5-15 are shown). The CESM2 and E3SMv2
simulations use the nominal iSSA perturbation designed to generate the target (-1.8Wm$^{-2}$) forcing when it is applied con-





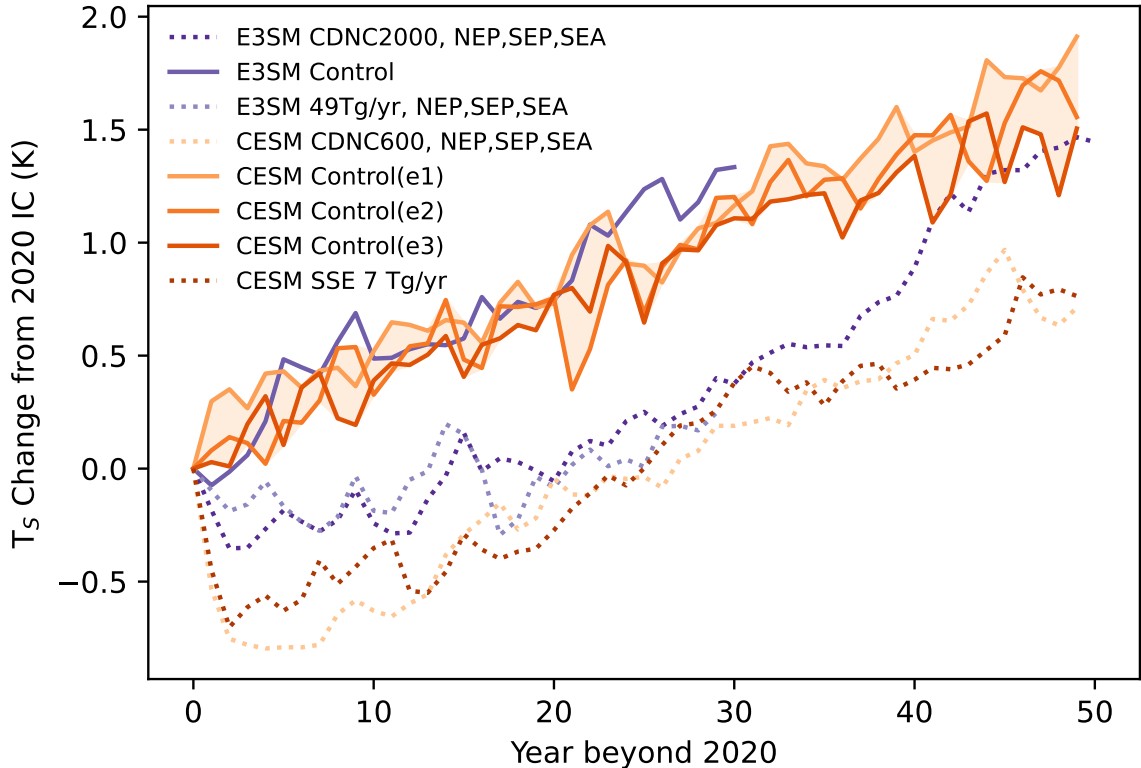

**Figure 6.** CESM2 and E3SMv2 time series of global averaged surface temperature change (with respect to the year 2000 initial conditions) for control (solid) and perturbation (dotted) simulations. CESM2 simulations are shown in purple, and E3SMv2 in orange shades. Three ensemble members for the CESM2 simulation are displayed to provide some sense of the model internal variability (the shading encompasses all three CESM2 control simulations).

currently in the NEP, SEP and SEA regions. The UKESM1 model used a somewhat stronger nominal emission magnitude of
50Tg/yr, which would produce a forcing of about -2.3Wm$^{-2}$(see upper panel of figure 3).

There are many precipitation responses common to all three models when the forcing is introduced in individual regions. The ITCZ generally shifts away from the hemisphere where MCB forcing is introduced, particularly over tropical oceans. Such a response to a cooling from radiative forcing is also found in idealized GCM studies (Kang et al., 2009). There is also a strong La Niña-like response to MCB forcing in the SEP and SP regions in all three models, with an eastward shift of warm
pool precipitation. All three models also show a precipitation reduction in eastern Amazonia when MCB is imposed in the SEA region, much like the opposite phase of an Atlantic El-Niño event that is observed in nature when warmer waters occupy the southeast Atlantic (Vallès-Casanova et al., 2020). The CESM2 model has a particularly large response compared to either E3SMv2 or UKESM1. All three models suggest that MCB forcing in the SEA region would introduce circulation changes that would be partially compensated for by MCB forcing in the SEP and SP region. It will be interesting to examine this
compensation in more detail in these simulations.



**Figure 7.** Precipitation responses produced through iSSA perturbations in coupled simulations, for three models with similar forcing (columns) when MCB is implemented in the regions indicated by red boxes (rows).





**Figure 8.** Surface temperature responses produced through iSSA perturbations in coupled simulations, for three models with similar forcing (columns) when MCB is implemented in the regions indicated by red boxes (rows).



Many of the features found in the surface temperature response are consistent with the precipitation response (Figure 8), consistent with Lindzen and Nigam (1987), but there are also important differences. Surface temperature changes over land differ between models (e.g., over Eastern China and much of N America) as do changes over the oceans around Australia, and at high latitudes. It is not clear whether these differences are associated with natural variability, and multiple realizations

(ensembles) will be important to assess the statistical significance of these results. We do not perform that kind of assessment here.

We find substantial differences in the GMST sensitivity to MCB depending on the intervention location and model, suggesting a substantial role for the feedback pattern effect. Notably, across the three models the SEA forcing produces weaker global cooling than other regions, while SEP and SP forcing shows stronger global cooling. Thus, the "global efficiency" of a given

MCB intervention depends not just on the susceptibility of the clouds to the MCB aerosol perturbations, but also the strength of the underlying radiative feedbacks operating with each region. The forcing produced by MCB in the NO region is substantially weaker than that produced by MCB imposed at lower latitudes because the area is smaller, the surface is often ice covered and the sunlight weaker, so induced changes in circulation features associated with precipitation and temperature changes are quite small compared to the impacts from forcing in other regions at these timescales, although there is some evidence for shifts

in the ITCZ along with the expected local Arctic cooling. Other signatures are likely to require longer simulations, or larger ensembles, since the Arctic climate (particularly sea-ice extent, and thickness) is extremely sensitive to small changes in the energy budget and terms contributing to those changes, and the Arctic is also a region of strong natural variability.

It is important to assess the utility of synthetic (SYN) estimates of climate responses (as described in Section 3.3.1 for fixed SST simulations) to make rough estimates of the climate response to particular combinations of forcings in multiple regions for

coupled simulations. This is a key assumption made when creating SRM controllers and large non-linearity would complicate MCB controller design. Using this methodology for coupled simulations is not as straightforward as when it is used to look at cloud and radiative responses in the fixed SST simulations because the different regions can interact much more strongly via thermodynamic and circulation changes when oceans are allowed to respond, more components of the climate system are allowed to interact, feedbacks operate, and some of the interactions operate on longer timescales. Some climate features

respond relatively slowly to radiative perturbations (on timescales of 30-50 years) and other features (e.g., polar features involve sea-ice and ice sheets) are governed by very delicate balances between processes, so accumulating impacts of short simulations is less reliable, and longer simulations, or large ensembles will be necessary.

In spite of a reliance on single realizations and short runs in many of our example figures, we think that synthetic estimates from shorter runs can provide good first order estimates of the response, and use of longer runs or ensembles will certainly

improve the situation. Figure 9 shows an example of the similarity in the surface temperature (TS) response in the synthetic estimate compared to that with MCB concurrently introduced in the NEP, SEP and SEA regions for years 10-30 in an SSP2-4.5 scenario using CESM2 and E3SMv2. The synthetic estimate corresponds quite closely to the response to concurrent forcing in terms of the general pattern and amplitude of the surface temperature change for CESM2. However, E3SMv2 shows significant non-additivity and the synthetic estimate cools more than the concurrent response by 0.3 K. The strong responses to the SEP





forcing seen in the tropical precipitation features of Figure 7 are also evident in the surface temperature response, and there is a noteworthy signal in the north Pacific where the Kuroshio Current contributes to the North Pacific Gyre as well.

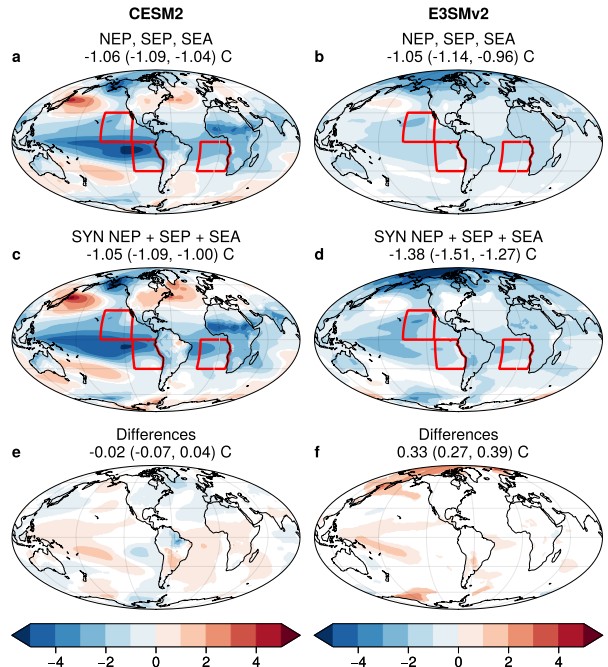

**Figure 9.** CESM2 7.5Tgy$^{-1}$ (left column) and E3SMv2 49Tgy$^{-1}$ (right column) 2m temperature response comparison between simulations in which iSSA are applied in the NEP, SEP, and SEA regions concurrent (top row) versus the synthetic estimate (SR123) computed by summing the response to each region individually (middle row). Red boxes display the emission regions. The bottom row displays difference between the top two rows (the non-additivity in the response). Grid points that are not statistically significant by the Student's t-test are masked in white. The mean GMST anomaly and the bootstrap resampled 5-95 percentile range are displayed in the titles of each panel.

To provide some evidence that synthetic estimates of the climate response are useful for other combinations of regions, Figures 10 and 11 show the estimated responses in surface temperature and precipitation when the three models are forced with iSSA emissions in the 4 pacific regions used in HEA2023 (and experiment E4 NEP+SEP+NP+SP of table 4). Both

concurrent and synthetic estimates are displayed. The CESM2 and E3SMv2 responses have been estimated using the standard procedure, by combining simulations made with nominal emissions confined to a single region, and comparing to a simulation made with the same emissions per region in all four regions concurrently (where all used constant emissions added to an SSP2-4.5 simulation). The methodology used for constructing the UKESM1 estimate differed because existing simulations from HEA2023 were exploited (their MCB simulations used a different background GHG scenario and they used time-varying

iSSA emissions). Details of the strategies used to generate the UKESM1 estimates shown in Figures 10 and 11 are provided in Appendix A2.





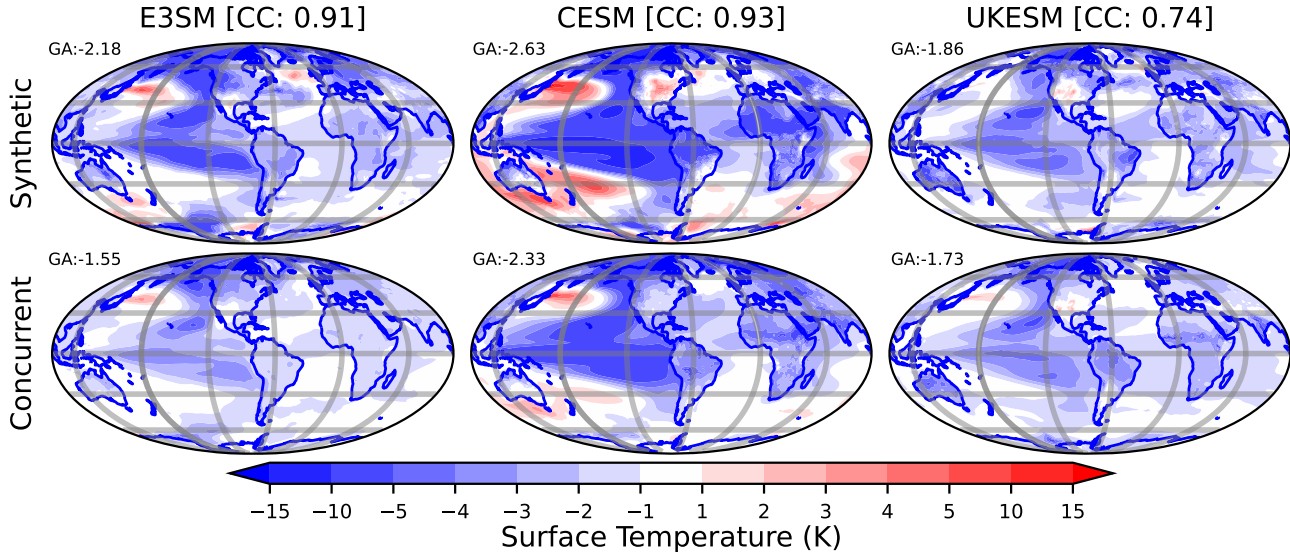

**Figure 10.** Surface temperature response to forcing in the NEP(R1), SEP(R2), SP(R4) and NP(R5) regions of CESM2, UKESM1 and E3SMv2 from iSSA emissions. The top row show the synthetic estimate and the bottom row shows the response when the forcing is induced concurrently. Area weighted Pearson Correlation Coefficients are shown in brackets following the model name in column titles. The UKESM synthetic estimate uses an SSP5-8.5 Scenario and averages decades 7 and 8 of model simulations for constructing the concurrent response estimate (see additional discussion in appendix A2)

The synthetic estimates of the responses generally produce a reasonable depiction of the responses to concurrent forcing, even when the estimate is produced with different GHG baselines as is the case with the UKESM1 model. When comparing the synthetic estimate of the climate response to the corresponding concurrent estimate it is clear that the magnitude of the
synthetic estimate is always stronger (for both temperature and precipitation) in all three models, but the large scale patterns match quite closely. Globally averaged temperatures differ by 0.6K, 0.3K, and 0.1K and correlation coefficients are 0.91, 0.93, and 0.74 in the E3SMv2, CESM2, and UKESM1 models respectively, with similar or better agreement in precipitation fields. There are also many small differences between the three models, some of which are likely due to natural variability that cannot be reduced without the use of ensembles. We have not evaluated ensembles and longer runs here, but they should play an
important role in a more systematic evaluation of models participating in an intercomparison. The La Niña-like response to forcing in the three region simulations noted above in the Figure 9 response is also present in the Pacific four region simulation but the response appears stronger (decreases in central Pacific precipitation and increases over northern Australia). All models show a weak temperature cooling (or a warming) in the North Atlantic and North Pacific suggestive of a strengthening of the Meridional Overturning Circulation resulting from MCB intervention. It is interesting to note that the patterns and amplitude of
the concurrent and synthetic estimates of the MCB responses for the UKESM1 model agree quite closely in spite of differences in the UKESM1 simulation setup, although the correlation are lower in that comparison than found in E3SM and CESM. So the details of the GHG forcing do not appear to matter a great deal to eliciting the MCB response, and even the details of the




MCB forcing may not be crucial in getting a rough idea of the response. CESM2 is also clearly the most sensitive to MCB forcing. It shows the largest response in the global averaged changes, and the amplitude of the anomalies (the positive and negative changes) in temperature and precipitation at the regional level are also much larger than for the other two models.

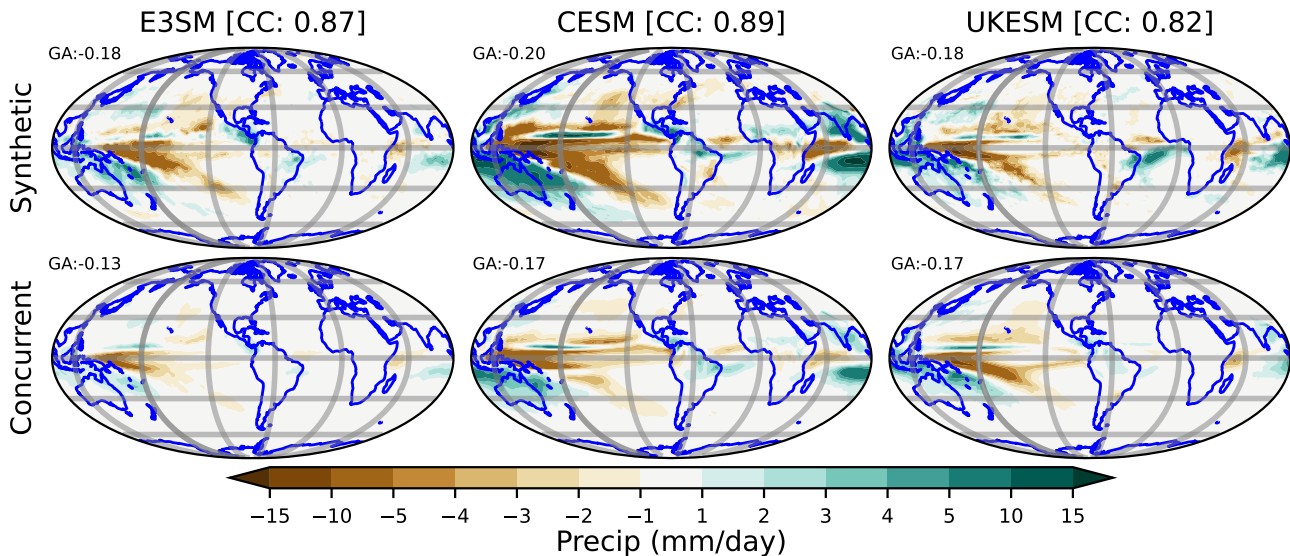

**Figure 11.** Precipitation responses to forcing in the NEP(R1), SEP(R2), SP(R4) and NP(R5) regions of CESM2, UKESM1 and E3SMv2 from iSSA emissions. Other details as in figure 10

## 4  Summary, next steps

This paper describes a modeling protocol designed to help understand climate impacts of Marine Cloud Brightening (MCB) Climate Intervention (CI). The simulations requested in the protocol are not intended to assess consequences from a realistic MCB implementation designed to minimize climate impacts over the globe, but instead to provide understanding about the kinds of responses produced by MCB interventions in particular regions, and provide rough estimates of the impacts from concurrent interventions in multiple regions. They are useful early steps to those more ambitious studies. Six regions are specified as candidates for MCB intervention in a simple, unambiguous way that can be easily implemented in models run with different resolutions and grid structures. The albedo change within regions is achieved by changing the in-cloud drop number concentration or by adding emissions of sea-salt aerosol in a size range believed to be optimal for MCB. A calibration step is employed to reach a common "nominal global MCB forcing" across models to allow differences between models to be assessed more easily, and to identify the relative importance of the forcing from cloudy and clear sky regions at forcing levels intended to counter a large fraction of the forcing from a doubling of GHG concentrations.

Once the calibration step is complete, coupled simulations with forcing in individual regions, and combinations of regions, are used to derive estimates of the climate impacts. "Synthetic" estimates constructed by linearly superposing responses from





simulations with forcing in individual regions can be calculated to approximate the climate impacts produced when MCB interventions are introduced in multiple regions. Examples were provided from simulations with 3 different climate models (CESM2, E3SMv2, and UKESM1), and those examples suggest that the synthetic estimates of the climate responses can be useful surrogates for responses in simulations when the intervention is introduced concurrently in multiple regions. We consider combinations of three and four regions in our example simulations. The most striking feature in terms of differences

in response is the high sensitivity of CESM2 clouds and the resulting radiative forcing to aerosol injection. This response in CESM2 is manifest through a large increase in the cloud fraction when compared to E3SM and UKESM1. While it might be tempting to view CESM2 as an outlier, there is gathering evidence from machine learning techniques applied to effusive volcanic eruptions that cloud fractions are strongly perturbed (Chen et al., 2022, 2024).Thus it is plausible that it is UKESM1 and ESMv2 that are inadequate in terms of the impact on radiative forcing. These results are very important in themselves as

they translate into emission rates that vary by around an order of magnitude which would have strong implications about the feasibility of MCB as a practical method of climate intervention.

We also explored how important the use of a common baseline scenario (e.g., the SSP scenario) is to an accurate estimation of the climate response. Our initial evaluation indicates that the choice of SSP is not critical, provided the evaluation is against control (baseline) simulations without an intervention and the evaluation is for a common interval (of years), in contrast to the

more frequent method of assessment comparing simulations with an intervention against a previous interval of years (often a "present day" or "pre-industrial" interval of years).

Our example assessments are simple, and somewhat superficial. They are intended to stimulate more study of these (and hopefully additional) simulations in order to provide deeper understanding. For example, our first look at the simulations indicate it would be useful to understand why the introduction of such different salt amounts is needed in different models, even

when two of the models (E3SMv2 and CESM2) share a common lineage for the cloud, aerosol, and radiation parameterizations (though the parameterizations have diverged in many ways). It would be interesting to explore the reasons for the disagreement in this relatively prescribed experimental setup. Comparisons with observations and LES simulations would also help, but reducing uncertainty and improving the representation of cloud aerosol processes sufficiently to inform better understanding of MCB is likely to also require expanded observations of the marine atmosphere and controlled studies of cloud aerosol

processes (e.g., Wood et al., 2017).

The models do appear to produce many common response signatures when forcing is introduced in these regions, supporting the robustness of these responses. It should be possible to use these simulations to improve understanding of climate responses to radiative forcing in these regions, and to guide more MCB research. A more rigorous and comprehensive community activity will require longer simulations, and more ensembles, in order to yield insights into climate impacts that manifest on longer

timescales, and to identify the role of natural variability in obscuring the signatures of the climate impacts and responses.

This protocol defines in detail the first two stages of an effort and lays the groundwork for subsequent stages. In these stages the same magnitude of MCB emissions (with similar mass/number per unit area) was used in any region that was active. After further understanding of responses is developed through the use of ensembles and longer simulations, and the robustness of the synthetic estimates of climate impacts are more fully quantified, next steps could examine whether different amplitude



emissions in different regions would be useful to optimize the climate response for different climate targets, and whether the same signatures (responses) occur across models. Initially it might be interesting to prescribe the amplitude of constant-in-time emissions in individual regions after an offline calculation is made to estimate the required amplitude in each region, to evaluate whether synthetic estimates of climate impacts are robust and whether simple "inverse" estimates of the requisite emission amplitudes are useful. Then a more sophisticated effort could use time-varying emissions, and involve either human or machine-based algorithms that vary the amplitude of intervention in time (i.e., the use of controllers) along with "peak shaving" intervention strategies similar to (Richter et al., 2022).

*Code and data availability.* E3SM source code for maintenance branch 2.0 may be accessed on the GitHub repository at https://github.com/E3SM-Project/E3SM and CESM2 at https://github.com/ESCOMP/CESM. Control simulations for CESM2, E3SMv2, and the UKESM are available from the Earth System Grid Federation nodes https://esgf.llnl.gov/. Details of how to access and run UKESM1 can be found at https://cms.ncas.ac.uk/unified-model/configurations/ukesm/relnotes-1.1/ (NCAS Computational Modelling Services, 2023). Source code modifications, model configuration information, simulation output for data appearing in this paper, and scripts used to produce the figures that are displayed in this study are provided via Zenodo at https://doi.org/10.5281/zenodo.10914383.

## Appendix A

### A1    Model Descriptions

#### A1.1    E3SMv2

E3SM is a fully coupled Earth system model developed by the U.S. Department of Energy (DOE) (Leung et al., 2020). Version 2 of E3SM (E3SMv2) has evolved significantly from earlier versions, especially in terms of fidelity measured by cloud and climate sensitivity metrics (Ma et al., 2022; Golaz et al., 2022).The E3SMv2 Atmosphere Model (EAMv2) has 72 vertical layers with a model top at approximately 60 km and horizontal resolution of 110 km. The E3SMv2 Land Model (ELMv2) has a horizontal resolution of 165 km. The ocean component (the Model for Prediction Across Scales-Ocean: MPAS-Ocean) in E3SMv2 has 60 vertical layers and its mesh spacing varies between 30 km (at the equator and poles) and 60 km (in the mid-latitdues). Aerosols in E3SMv2 are simulated by the four-mode version of Modal Aerosol Module (MAM4) (Liu et al., 2016; Wang et al., 2020). Among the major aerosol components represented in MAM4, sea salt aerosol is represented in the Aitken, accumulation, and coarse mode with particle emission size (diameter) ranges of 0.02-0.08, 0.08-1.0, and 1.0-10.0 $\mu$m, respectively. The emission flux of natural sea salt is determined by first dividing the particle distribution into 31 size bins, following the parameterization of Mårtensson et al. (2003) and Monahan et al. (1986), and then redistributing the emissions into the three MAM4 size modes. In the E3SMv2 MCB experiments for this study, we emit the additional sea salt particles into size bins with a diameter of 0.082 and 0.104 $\mu$m that lead to an increase of sea salt emissions primarily in the accumulation mode.





### A1.2   CESM2

CESM2.1 is a fully coupled Earth system model developed by the University Corporation for Atmospheric Research, National Center for Atmospheric Research, and other community members. CESM2 is described in Danabasoglu et al. (2020) and the production release version 2.1.4 is used here. The atmosphere component is the "low-top" Community Atmosphere Model 6 configuration which has 32 vertical layers with a model top at approximately 40km and we use the finite volume dycore with a horizontal resolution of 0.9° latitude by 1.25° longitude. The land component is the Community Land Model 5 also at 0.9 by 1.25 resolution. The ocean component is the Parallel Ocean Program version 2 (POP), using the displaced Greenland pole 1-degree grid. Aerosols in CESM2.1 are simulations are also simulated by MAM4 and the sea salt aerosol emission parameterization is the same as in E3SMv2. Similarly, additional sea salt particles are emitted into size bins with dry diameters 0.082 and 0.104 $\mu$m.

### A1.3   UKESM1

UKESM1 is a fully coupled Earth-system model developed jointly by the UK's Met Office and Natural Environment Research Council. The model is described by Sellar et al. (2019) and was used to deliver simulations for the Coupled Model Intercomparison Project Phase 6 (Eyring et al., 2016). The atmosphere model component (Walters et al., 2019) has 85 levels with a model top at approximately 85 km and horizontal resolution of 1.25° in latitude by 1.875° in longitude. This is coupled to an ocean model (Storkey et al., 2018) of 75 levels and 1° resolution. Other components simulate tropospheric and stratospheric chemistry (Archibald et al., 2020), vegetation and the land surface (Best et al., 2011), ocean biogeochemistry (Yool et al., 2013) and sea-ice (Ridley et al., 2018). Aerosols are simulated by the GLOMAP-mode scheme (Mann et al., 2010) using five log-normal modes for sulfate, sea-salt, black and organic carbon, while a sectional scheme is used to simulate mineral dust (Sellar et al., 2019).

### A2   Info on use of the UKESM1 simulations

This section provides a little extra detail about how simulations from HEA2023 were used in this study. As with CESM2 and E3SM, in HEA2023 a set of stage 1 calibration runs and single region simulations to estimate forcing were performed. The results of forcing produced by varying emissions in each region are shown in table 4. Coupled single region simulations at a nominal emission rate were then made in a similar manner to experiments E4-XXX of table 4 (where XXX represents the region abbreviation for the 6 different regions) were then performed with an SSP2-4.5 scenario as its baseline. But HEA2023 chose a constant (50 Tg/yr/region) iSSA emission rate for their nominal emissions. That emission rate would be considered "on the high side" if the guiding intent had been to produce a -1.8 Wm$^{-2}$ forcing for the R1+R2+R3 region. The two rightmost columns of table A1 show the synthetic estimates of forcing for the 3 and 4 regions considered for this study. The synthetic estimates suggest that a 25Tg/yr/region emission rate would produce a forcing closest to our target "nominal forcing". Emissions at 50Tg/yr/region would produce a forcing of approximately -2.42 Wm$^{-2}$ if used over 3 regions, and -3Wm$^{-2}$if used in the 4 region area.



**Table A1.** MCB Forcing produced by varying emission rates (per region) with emissions in single regions in fixed SST simulations (W/m2). The rightmost two columns display synthetic estimates of the forcing that would be produced by emissions in the NEP+SEP+SEA and NEP+SEP+SO+NO (labelled S123 and S1245 respectively)

| Em Rate | R1 | R2 | R3 | R4 | R5 | | |
|---|---|---|---|---|---|---|---|
| Tg/yr | NEP | SEP | SEA | NP | SP | S123 | S1245 |
| 0 | 0 | 0 | 0 | 0 | 0 | 0 | 0 |
| 5 | -0.08 | -0.26 | -0.09 | -0.24 | -0.23 | -0.43 | -0.81 |
| 12.5 | -0.15 | -0.41 | -0.32 | -0.33 | -0.38 | -0.87 | -1.27 |
| 25 | -0.43 | -0.60 | -0.52 | -0.46 | -0.57 | -1.55 | -2.06 |
| 50 | -0.59 | -0.93 | -0.90 | -0.59 | -0.85 | -2.42 | -2.96 |
| 100 | -0.97 | -1.49 | -1.36 | -0.85 | -1.02 | -3.82 | -4.33 |

The only coupled simulations performed in HEA2023 that generated a climate perturbation by applying concurrent emissions in multiple areas employed time varying iSSA emissions and MCB forcing introduced on top of an SSP5-8.5 scenario. The iSSA emissions were adjusted manually each decade during the simulations to make the global averaged surface temperature agree approximately with the global averaged temperature following an SSP2-4.5 scenario. In this way time varying MCB emissions were used to partially compensate time varying GHG forcing. Figures A1 and A2 display the emissions in the pacific regions (SEP, NEP, SP, and NP) needed to achieve the required cooling. Figure 2 and table A1 indicates that to achieve a -1.8W/m2 forcing globally about 100 Tg/yr would be needed (or about 25 Tg/yr per region) and figure A2 indicates this occurs during the decade starting in 2071.

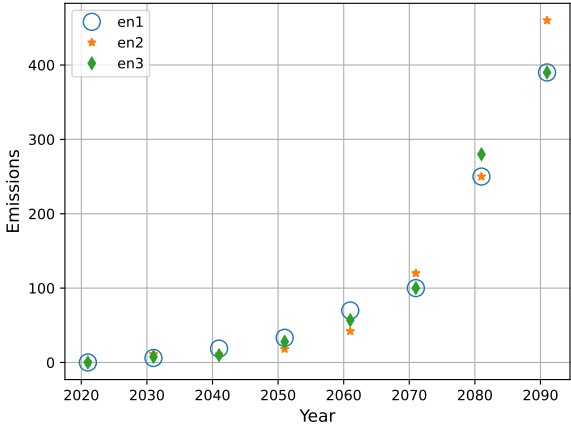

**Figure A1.** UKESM1 Concurrent emissions per decade in regions SEP, NEP, SP, and NP.

Because of these choices both the GHG and MCB forcing amplitudes differed compared to simulations and results produced with the other two models. Synthetic estimates of the climate response were produced by summing the changes in climate features from the simulations with four individual regions with 50 Tg/yr/region emissions, and then compared to UKESM1 simulation changes averaged over decades 7 and 8 of the G6MCB simulations (which also emitted approximately 50Tg/re-



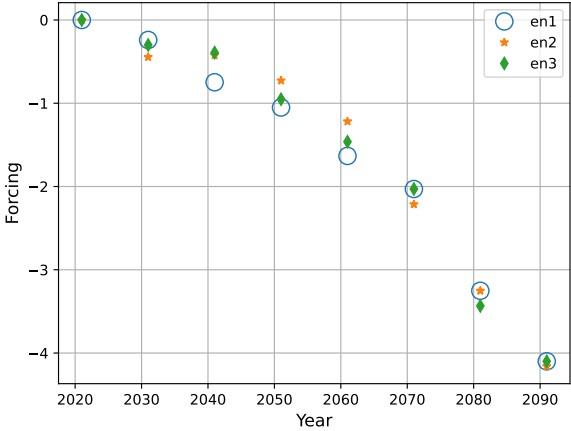

**Figure A2.** UKESM1 synthetic estimates of Forcing per decade for concurrent emissions in regions SEP, NEP, SP, and NP.

gion/year when averaged over those two decades). Since emissions at this amplitude in the 4 regions produce a stronger
forcing than used in the E3SM and CESM simulations one might anticipate that the climate response would also be somewhat larger. We use these strategies to displayed the estimates for Surface Temperature and precipitation changes in figure 10 and 11.

## A3   Recommended Output

We recommend participating models provide data atmosphere, ocean, and land variables required for diagnosis of climate
impacts and aerosol-cloud interactions. The Climate Model Intercomparison Project (CMIP) tier-1 variables provide a starting point for the data request (https://cmip6dr.github.io/Data_Request_Home/ and https://github.com/cmip6dr/data_request_snapshots/blob/main/Release/dreqPy/docs/CMIP6_MIP_tables.xlsx). We request that the datasets follow the Climate and Forecast (CF) Metadata conventions (https://cfconventions.org/) but do not require the data be converted to the more restrictive CMIP variable naming conventions or use the climate model output rewriter (CMOR, see, e.g., https://cmor.llnl.gov/). Model
output can be provided in the netCDF or zarr file formats.

*Author contributions.* PJR wrote first draft of protocol, designed simulations, created many figures, and drafted the paper. All authors commented on and improved the first draft of protocol, and the manuscript. HH, MW, and AJ carried out simulations, created some figures, and wrote parts of the paper. SD, RW, HW and JH co-designed the simulations and helped with interpretation of results. HH and MW created the time series of CESM2 and E3SMv2 dataset, and archived all the data. HS gave input to simulation design.

*Competing interests.* The contact author has declared that none of the authors has any competing interests.



**Figure A3.** Comparison of 2m air temperature response to R1+R2+R3 MCB using CDNC (top row) versus iSSA (middle row) and their difference (bottom row) for the 2025-2044 anomaly from SSP2-4.5. For CESM2 (left column) we compare CDNC = 600cm$^{-3}$ to iSSA = 7.5Tgyr$^{-1}$ and for E3SMv2 we compare CDNC = 2000cm$^{-3}$ to iSSA = 49Tgyr$^{-1}$, which all produce similar ERF magnitudes. A red dashed contour denotes the -10Wm$^{-2}$ contour of the top of atmosphere downward radiative flux anomaly from the associated fixed SST simulation for each simulation, to indicate the slight differences in the pattern of forcing between CDNC and iSSA simulations. Masked grid points denotes that are not significant by the Student's t-test and false detection rate. Bootstrap resampled mean and 5-95 percentile range of global mean surface temperature anomalies are shown above each panel.





**Figure A4.** As in Fig. A3 but for precipitation (mm/day).





**Figure A5.** Comparison of CESM2 CDNC=600cm$^{-3}$ (left column) and E3SMv2 CDNC=2000cm$^{-3}$ (right column) R1+R2+R3 simultaneous forcing response in the early 21st century (2025-2034; top row) versus mid century (2055-2064; middle row) and their difference (bottom row). Hatching denotes grid points that are not significant by the Student's t-test and false detection rate. Bootstrap resampled mean and 5-95 percentile range of global mean surface temperature anomalies are shown above each panel.



*Acknowledgements.* Support for Philip Rasch, Haruki Hirasawa, Sarah Doherty and Rob Wood was provided by SilverLining's Safe Climate Research Initiative (SCRI) and through the University of Washington's Marine Cloud Brightening Program, which is funded by the generous
support of a growing consortium of individual and foundation donors. Andy Jones and Jim Haywood were also supported by Silverlining's SCRI and the Met Office Hadley Centre Climate Programme funded by DSIT. Mingxuan Wu and Hailong Wang were supported by Silverlining's SCRI subcontract to PNNL. This publication is partially funded by the Cooperative Institute for Climate, Ocean, & Ecosystem Studies (CICOES) under NOAA Cooperative Agreement NA20OAR4320271, Contribution No. 2024-1351. Computational resources were provided by Amazon Web Services.
Thanks to Linda Hedges (Silverlining), and Brian Dobbins (NCAR) for AWS technical support. We thank Jean-François Lamarque for discussion and comments about the protocol and paper contents.

The Community Earth System Model (CESM) project is supported primarily by the National Science Foundation. The E3SM project is supported by the U.S. Department of Energy (DOE), Office of Science, Office of Biological and Environmental Research. The Pacific Northwest National Laboratory (PNNL) is operated for DOE by Battelle Memorial Institute under contract DE-AC05-76RLO1830.



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
