# Peer review of "A protocol for model intercomparison of impacts of Marine Cloud Brightening Climate Intervention"

_EGUsphere, 2024_

## Author Response (AR1)

Our responses to RC1(Ben Kravitz), RC2, and RC3 appear on the following pages

Phil Rasch

Responses to Referee Comments GMD egusphere-2024-1031

Referee Comments in blue, our responses in black

Response to Ben Kravitz:

This paper is superb. When the authors present their ideas for a modeling protocol, one can tell that it's based on a huge amount of experience in modeling aerosols and clouds. This is very well thought through. Separating the protocol into two stages is an excellent way of separating inter-model and process uncertainties. I particularly appreciate the clear, approachable writing style. I have a few comments, mostly minor.

Thank you for the kind words and constructive comments

The abstract is a bit vague. I was hoping to see more description of actual results or details.

We have rewritten the 2nd paragraph to provide some results and details.

Section 1 is a superb introduction to the subject.

Thanks

Lines 137-138: You could cite Visioni et al. (2024) – the most recent G6 simulation protocol.

Done

Section 2.1 (or elsewhere as you see fit): Some new work by Jack Chen (doi:10.1029/ 2024GL108860, should be available in early review any day now) around seeding regions other than the most susceptible are interesting from a teleconnections perspective.

We have modified the discussion in section 2.1 to account for Jack's new work. We have added a few words to some sentences pointing out that the strong regional responses are often associated with forcing in the subtropical stratocumulus regions, and also now added sentences to cite Jack's study indicating that those strong regional response are not necessarily produced when forcing is introduced over a larger aerial extent in less susceptible marine clouds.

Lines 212ff: Yes, more reliable, but there are still parameterizations that result in substantial differences. See various work involving M. Ovchinnikov. There's also the issue that LES is often run with a doubly-periodic domain, which does have effects.

We added a sentence at the paragraph end providing a few caveats about LES models.

Lines 227-228: I think this is highly sensible and uses ESMs to their full advantage.

Table 1, Q1:  I would suggest cutting the part about "recent climate models".  Your protocol would work for older, "less sophisticated" models, and exploring that could be really interesting for understanding climate responses.

We agree less sophisticated models could also produce useful results. The question was intended to identify whether the improvements  in modern climate models (in scenarios, in experimental design, and in aerosol and cloud microphysics) generate different results from older models and studies. We added some words to Q1, and modified the sentences in the text discussing the question, and hope the revisions clarify our intent.

Table 1, Q3:  This is phrased like a threshold, which is less powerful than it could be.  I might suggest "What is the relationship between seeding area and forcing?"  And then you could fold Q7 into this one.

Thanks for the suggestions. We understand your point, but think your suggested rephrasing misses some of the nuance we were looking for. We have tried to rephrase Q3 to capture our intent better (again making a longer sentence unfortunately).  We hope it is now distinct from Q7, which remains unaltered to emphasize the importance of additivity and linearity.

Table 1, Q4:  I'm not entirely sure I understand the last clause.

 We have added necessary words to clarify the clause (we hope!).

Table 1, Q5:  And are these model-dependent?

 We added your words to the end of the sentence :-)

Lines 282ff:  Jack Chen has done some work on this (see comment above, and there are other recent papers).  It would be worth (somewhere) commenting on whether his approach is a sensible idea for what you were thinking.

We think Jack's work is very interesting and provides food for thought. We don't think it is enough to guide the development of a controller for MCB yet, but know it will have an influence on strategies for controller development. We added some discussion of these issues in the last paragraph of section 4, where we discuss next steps of the protocol.

Lines 299ff:  I'd be careful with this.  If you're endorsing slab ocean simulations, you will need to specify what the ocean fluxes should be (preindustrial, "present day", SSP, etc.).

We decided to remove the discussion of slab ocean models.

Lines 320ff:  How many models can actually do this?  Can activation schemes give the right answer?

We are not sure whether you really want statistics, or if this question is rhetorical. Our target audience is the modeling community capable of handling clouds, cloud feedbacks, and aerosol-cloud interactions (ACI). I (PJR) think most credible ESMs participating in CMIP/IPCC can do clouds and cloud feedback, so they could do the CDNC simulations. A smaller fraction can do ACI, with varying degrees of sophistication about how to handle aerosol activation. But these numbers are purely based upon intuition, and I am not sure how to make such an estimate rigorous. If some group wants to invent a different method to introduce a forcing in the protocol regions they could certainly participate on the stage 2 simulations but the protocol isn't designed for them, would defeat some of the purpose of establishing the uncertainties in the GCM cloud behavior, and I wouldn't know how to compare their results with much rigor.

There are now numerous treatments of aerosol activation, and our 3 example models use the very popular (but simple) Abdul-Razzak and Ghan treatment. We do have more sophisticated (and accurate) activation schemes available in all 3 models which are sometimes used (with increasing frequency in UKESM).  We have chosen not to modify this section because we believe you don't expect a "yes" or "no" answer to this question, which requires a dedicated comparison of different activation schemes. Regardless, we don't expect models to have identical sea salt injection rates, size distribution, CDNC and cloud forcing responses to the injections.

Section 3.1:  I know you're allowing models to include whatever parameterizations they have, but I think it would be worth a comment on mixed-phase clouds in the Northern Oceans.  That regime is quite different from subtropical cloud decks.

We have added a few sentences to the paragraph beginning near (old version) line 337 commenting on the complexity of  mixed phase clouds and now provide a reference.

Line 347:  As written, this is confusing.  Just say -1.8 W/m2 here and provide a tolerance (±0.1 W/m2?).

Done

Table 4 makes a lot of sense, but it's a big lift for some modeling groups.  Is there a way you can provide a prioritization for interested groups?

We have added sentences after introducing Table 4 to help participants prioritize the simulations.

Lines 393-394:  Can "substantial fraction" and "almost entirely" be numbers?

We replaced words with numbers :-)

Line 394: I'm confused by "a factor of 10 increase". Is that an increase over baseline (and if so, what are the base values)? Or an increase over CESM?

We changed the sentence to be clearer.

Lines 420-421: These results are fascinating. If they hold for other models, how might that modify your simulation protocol? Does that help you deprioritize certain experiments?

We don't think this result is a surprise, because the aerosol lifetime is short, and the (cloudy and clearsky) forcing discussed on those lines is calculated using fixed SST simulations. The fixed SST runs are relatively short and cheap so I (PJR) feel it is useful to do all the stage 1 simulations for all models, just to make sure that there are no surprises. Therefore we do recommend that all the fixed SST runs are performed, and do not recommend deprioritizing any of the stage 1 simulations. We also assigned priorities to some of the coupled simulations.

Lines 429-430: I don't understand how this follows from the results described in this paragraph. Or did you mean this sentence to apply to the previous paragraph?

Thanks for noticing inconsistencies here. Both figure 3 and 4 reported results from stage 1 simulations with forcing in individual and concurrent regions (Figure 3 shows patterns for forcing in three regions, Figure 4 summarizes global averages for all independent regions, along with estimates for 3 and 4 region combinations). A small amount of the information was redundant, and confusing. We have revised the text, and hope it is now clearer.

Figure 6: What happens in E3SM after 30 years?

We simply terminated the simulation, so we didn't have any more years to plot. But we extended the R1-3 simulations to 50 years and the SSP2-4.5 to 70 years, and have now updated the figure with the updated time series.

Lines 489-490: I think you can cut this sentence.

Done

Figures 7 and 8: The results are remarkably robust across models (except for CESM2's super-ENSO). This is promising for a larger coordinated effort.

Agreed.

Line 491: I think your figure referencing/numbering is off.

Fixed. Thanks.

Lines 510-511:  These nonlinearities are important for design, but it's not an assumption.  We do test this.

Agreed… changed the sentence  to say "large non-linearity will play a role in MCB controller design".

Line 515:  Responses on long timescales do not affect the operation of the controller, although they are relevant for what you're talking about.

Agreed… changed the phrase "accumulating impacts of" to "identifying impacts during"

Line 519:  Citation to Wan et al. (2014) might be relevant?

Agree that Wan et al  could be a relevant study (if my assumption that you are referring to Hui Wan's 2014 paper) but to keep the number of references under control, we have listed the previously cited papers by Kay, and Tilmes which take a more traditional approach to the use of ensembles.

Figures 10 and 11:  Can you add a row showing difference plots?  It's hard to eyeball.

If you and the editor will allow it, we hope to avoid showing the difference plots. We believe the correlation coefficients, and the mean values provide enough info about the similarities and differences to get a sense for the methodology. Our intention was merely to provide an indication that patterns and amplitudes of synthetic and concurrent circulation responses were similar as an illustrative example (as stated in the section 3.3 title and first sentence).  If one wants to be more quantitative, then we feel it would be better to use longer runs and/or ensembles, which is a topic planned for a future study but beyond scope for this paper.

Appendix A3:  It might be worth commenting on some specific variables, such as the importance of clear sky forcing values.

Done

Responses to Referee Comments GMD egusphere-2024-1031

Referee Comments in blue, our responses in black

Response to RC2

general comments

The manuscript of egusphere-2024-1031) presented a protocol for model intercomparison of marine cloud brightening climate intervention. The target is important and the manuscript is generally well-written. I agree with the publication of this manuscript, but I have the following comments. Please clarify them.

major comments

The manuscript contains so many abbreviations. We may understand through the manuscript; however, how about providing the list of abbreviations used in this manuscript as a table (or appendix table)?
I understand that this manuscript presents an example of the protocol's outline, but why three climate models (E3SMv2, CESM2, and UKESM1) were selected as an example? Did it simply because one (or more) of co-author(s) conduct model simulation? Or, the authors expected the variability (i.e., large difference) of results among three models beforehand? This kind of information may encourage readers to join this protocol.

Thanks for the constructive comments.

We have added a table of acronyms as  appendix A4 (to avoid renumbering and now mention it in the last sentence of the introduction.

The 3 models were merely for convenience because we were working together, and had expertise with those models. We added a sentence to section 3.3 stating this information.

specific comments

P4, Line 92-93: This sentence could be moved to the last part of Section 2. Please reconsider.

Done

P6, Line 167: The explicit definition like "Sea-Salt Aerosol (SSA)" seems to be better.

We originally did choose the acronym SSA for the aerosol to be injected for MCB intervention, but we were concerned because people often assume that aerosols called sea-salt (or

sea-spray) can consist of particles with a size distribution similar to sea-spray from a natural source, and as discussed in the manuscript, both radiative and microphysical characteristics of the smaller particles with a narrow size distribution optimized for MCB can be quite different from those from a natural source. So we deliberately chose to use a non-standard identifier. Apparently our explanation for the naming convention can still be improved. So a few more words have been added to the paragraph in the hope it helps.

P8, Table 1: Need period for this caption.

Done.

P9, Table 2: This caption needs more description even though this can be "see text  for more discussion". Several sentences need a period.

Done.

P13, Line 378: At this point listing Table 4, the reason discussed in line 297-291 can be raised again.

We assume you meant lines 287-291. We have repeated some of that information and revised the text to make it read more smoothly and logically.

P14, Table 4: From the current description, I do not fully follow the description of "30 to 80 yr simulation" in stage 2. Does it allow at least 30 yr simulation as experiments? What is the reason for setting the maximum length as 80 years?

The 80 year simulation was chosen to extend the simulations to year 2100, a stopping point frequently chosen in model intercomparison studies, e.g. SSP2-4.5 which were our baseline simulations (although many signatures are available from a shorter range of years). The long simulation range is pretty arbitrary, but was chosen to provide more opportunity for evaluating longer timescale model responses, and more information about impacts of MCB on multi-decadal variability.

P15, Line 390-394: I agree with these discussions, but the discussion in terms of Fig. 2 will also require the difference between all sky and clear sky. Compared to the clear sky condition, the difference among the three models (CESM2, E3SMv2, and UKESM1) under all sky conditions seems to be large.

Ben Kravitz also asked for a little more information about partitioning between cloudy and clear sky forcing for each model. We have added additional information about this in our discussion of Figure 2 and Figure 5 to respond to both of you.

P17, Table 5: The names of the models should be unified throughout this manuscript.

Done.

Done.

We have revised Figure 9 and now indicate statistically insignificant regions using hatching
rather than color.

We have redrafted figures with appropriate model versions, and we now explain the
abbreviation GA.

technical corrections

Lines 198, 222, 267, 293, 344, 379, and 441: Section titles such as "Strengths", "Protocol
Design", "Protocol", "Setup, Experiments:", "Procedure", "Sea Surface Temperatures" and
"Simulations" included capital letters. Please revise.

Done.

The unit should be unified (using / or -; please also check the format guideline of this journal).
For example, Line 389 (Figure 2) and relevant discussion parts used "Tg/yr" in the x-axis
whereas "Wm-2" in the y-axis (In this case, please also use superscript for m2). In addition, "y"
or "yr" should be unified throughout this manuscript.
Throughout the manuscript, space is needed between the digit and the unit. Please confirm in
the revision process.

We have modified the conventions for units in all of the figures.

Responses to Referee Comments GMD egusphere-2024-1031

Referee Comments in blue, our responses in black

Response to RC3

Current Marine Cloud Brightening (MCB) type solar radiation modification modeling activities, especially those within the GeoMIP framework, often employ highly simplified representation of MCB. This approach suggests that the climate response may significantly diverge from that of a realistic MCB implementation. The manuscript introduces an MCB modeling intercomparison protocol that expands upon prior MCB modeling research. It aims to deliver a more systematic and coherent framework for evaluating the climate impacts of various MCB strategies and for quantifying the climate response to specific MCB forcings across different regions. This novel protocol would help to address many questions that have arising from prior MCB studies, as outlined in Table 1 of the manuscript, and to deepen our understanding of MCB through its hierarchical design. It is recommended that the proposed MCB modeling protocol be integrated into the forthcoming GeoMIP for CMIP7, thereby inviting more participation from modeling teams and extending benefits to the broader geoengineering community.

Broadly, the manuscript provides a comprehensive exposition of the experimental design, the calibration procedures, and presents simulation outcomes for three coupled climate models. The manuscript is clear and well-composed. Despite the presence of a few minor typographical errors, they are expected to be remedied during the proofreading phase.

Thanks for the comments.  The reviewer recommends that the protocol be proposed for GeoMIP and CMIP7. We hope that it is discussed and considered at the upcoming GeoMIP meeting, (possibly with some extensions that will be discussed by co-authors attending the meeting). For convenience we have decided to name the protocol MCB-REG and refer to it that way hoping a name will facilitate discussions also. The reviewer also noted typos. We have now corrected many typos and addressed a number of minor issues raised by the other two reviewers.

---

## Author Response (AR2)

Responses to Referee Comments GMD egusphere-2024-1031

Referee Comments in blue, our responses in black

Response to RC3 comments on Revision #2

This manuscript is the revised version that publicly reviewed in the egusphere. I appreciate the authors' responses to address my concerns. Most of them have been clarified well, but I would like to request the following minor revisions before the publication process.

Thanks!

P10, Table 2: The caption is not still informative. Please add a short description of this table.

We changed the table caption to provide more information

P22, Figure 6: In the x-axis, what is "(CE)" as unit? In y-axis, "(C)" should be "(K)"? Please confirm these units.

CE acronym stands for "Common Era". It is a newer term, but it was not necessary, so we have removed it. We have change "C" to "K"

P26, Figure 9: The unit of "TREFHT" has not been defined. Please clarify.

We have changed the "C" appearing in the panel heading above the figure with "K", added "K" below the colorbar, and specify the units in the figure caption.

---

## Author Response (AR3)

Sept 13, 2024

I am submitting a revised manuscript to account for the following editor and copy editor suggestions:

1. The latest version of the manuscript now uses only black text (copy editor request)
2. Figure 3 and 9 are now displayed in a larger size in accordance with the editors request.
3. The fonts have been made larger in Figure 5  to make the figure easier to read as requested by the editor.

Thanks

Phil Rasch

---

## Author Response (AR4)

The content of the manuscript is unchanged since the last version.